# Construction of the axolotl cell landscape using combinatorial hybridization sequencing at single-cell resolution

Fang Ye[1,2,10], Guodong Zhang[1,10], Weigao E.[1,10], Haide Chen[1,10], Chengxuan Yu[1,10], Lei Yang[1,10], Yuting Fu[1], Jiaqi Li[1], Sulei Fu[3], Zhongyi Sun[1], Lijiang Fei[1], Qile Guo [4], Jingjing Wang[1,2], Yanyu Xiao[1], Xinru Wang[1], Peijing Zhang[1,2], Lifeng Ma[1], Dapeng Ge[1], Suhong Xu [1], Juan Caballero-Pérez [5,9], Alfredo Cruz-Ramírez[5], Yincong Zhou[6], Ming Chen [6], Ji-Feng Fei [3✉], Xiaoping Han [1,7✉] & Guoji Guo[1,2,4,7,8✉]

The Mexican axolotl (*Ambystoma mexicanum*) is a well-established tetrapod model for regeneration and developmental studies. Remarkably, neotenic axolotls may undergo metamorphosis, a process that triggers many dramatic changes in diverse organs, accompanied by gradually decline of their regeneration capacity and lifespan. However, the molecular regulation and cellular changes in neotenic and metamorphosed axolotls are still poorly investigated. Here, we develop a single-cell sequencing method based on combinatorial hybridization to generate a tissue-based transcriptomic landscape of the neotenic and metamorphosed axolotls. We perform gene expression profiling of over 1 million single cells across 19 tissues to construct the first adult axolotl cell landscape. Comparison of single-cell transcriptomes between the tissues of neotenic and metamorphosed axolotls reveal the heterogeneity of non-immune parenchymal cells in different tissues and established their regulatory network. Furthermore, we describe dynamic gene expression patterns during limb development in neotenic axolotls. This system-level single-cell analysis of molecular characteristics in neotenic and metamorphosed axolotls, serves as a resource to explore the molecular identity of the axolotl and facilitates better understanding of metamorphosis.

[1] Center for Stem Cell and Regenerative Medicine, and Bone Marrow Transplantation Center of the First Affiliated Hospital, Zhejiang University School of Medicine, Hangzhou 310000, China. [2] Liangzhu Laboratory, Zhejiang University Medical Center, Hangzhou, Zhejiang 311121, China. [3] Department of Pathology, Guangdong Provincial People's Hospital, Guangdong Academy of Medical Sciences, Guangzhou 510080, China. [4] Zhejiang University-University of Edinburgh Institute, Zhejiang University School of Medicine, Zhejiang University, Hangzhou 314400, China. [5] Molecular and Developmental Complexity Group, Unidad de Genómica Avanzada, Laboratorio Nacional de Genómica para la Biodiversidad, Cinvestav Unidad Irapuato, Km. 9.6 Libramiento Norte Carretera. Irapuato-León, 36821 Irapuato, Guanajuato, Mexico. [6] College of Life Sciences, Zhejiang University, Hangzhou 310003, China. [7] Zhejiang Provincial Key Lab for Tissue Engineering and Regenerative Medicine, Dr. Li Dak Sum & Yip Yio Chin Center for Stem Cell and Regenerative Medicine, Hangzhou, Zhejiang 310058, China. [8] Alibaba-Zhejiang University Joint Research Center of Future Digital Healthcare, Hangzhou, Zhejiang 310058, China. [9] Present address: EMBL-EBI, Wellcome Genome Campus, Hinxton, Cambridgeshire CB10 1SD, UK. [10] These authors contributed equally: Fang Ye, Guodong Zhang, Weigao E, Haide Chen, Chengxuan Yu, Lei Yang. ✉email: jifengfei@gdph.org.cn; xhan@zju.edu.cn; ggj@zju.edu.cn

Cell proliferation and differentiation occur actively during development and decline dramatically in adulthood in a multicellular organism. Regeneration of a complex organ, similar to developmental process, requires coordination of the proliferation and differentiation of multiple cell types, and such ability is restricted in most mammals. The Mexican axolotl (the salamander *Ambystoma mexicanum*), however, exhibits strong inherent regeneration capacity in many complex body structures including limbs, tail, eye, heart, lung, gill, and central nervous system[1]. As an important tetrapod model in organ development and regeneration studies, the axolotl is one of the keys to explore the regeneration of mammals. Activating the potential regeneration capacity of the human body would be valuable in regenerative medicine[2].

Axolotls do not metamorphose naturally, but retain the morphology of the larval stage after limb development as sex-mature status, which is known as neoteny[3]. Neoteny is an intriguing developmental fate that is present to some degree in various vertebrates, such as naked mole-rats (*Heterocephalus glaber*), axolotls and humans[4]. Neotenic axolotls show resistance to aging-related phenotypes and diseases[5]. With a reduced water level or the administration of thyroid hormones (thyroxine, T4), neotenic axolotls can irreversibly transform into terrestrial salamanders that lack fins and external gills[6,7]. The metamorphosis of adult axolotls eventually results in a severely reduced regeneration ability[8]. The relatively long lifespan of neotenic axolotls is partially attributed to their extraordinary regenerative capacity[9,10]. Previous studies have identified the endocrinological axis and related tissues that control metamorphosis[11,12]. The hypothalamic–pituitary–thyroid axis serves as a regulatory system for thyroxine conversion. A lack of 3,5,3′-triiodothyronine (T3) receptors and a low level of transformed thyroxine lead to neoteny in the axolotl. Gene expression programs activated by thyroxine during axolotl metamorphosis remodel a fair proportion of organs in this organism, and some genes involved in the response to thyroxine have been identified using the subtractive hybridization method[13].

Recent advances in large-genome assembly and tissue-mapped de novo transcriptomics in axolotls have enabled further surveillance of gene expression patterns at the single-cell level[14–17]. Limb regeneration in neotenic or metamorphosed axolotls has been characterized using single-cell transcriptome sequencing and proteomic analysis[18–23]. But the systematic cell composition and interaction landscape of axolotls remain to be solved. On the other hand, although transcriptional perturbations during thyroxine-induced metamorphosis in the axolotl have been evaluated by qRT–PCR and multiorgan bulk RNA-seq[24,25], the cellular heterogeneity of remodeling organs in metamorphosed axolotls is still unclear. Considering the metamorphosis as a fascinating biological process, and its association to the regeneration and life span, it will be important to investigate the molecular and cellular changes occurred in metamorphosis, i.e., comparing systematically the alterations before and after metamorphosis.

In recent years, high-throughput single-cell transcriptomics and chromatin accessibility methods have generated numerous resources and provided new insights into the cell heterogeneity of tissues or whole organisms[26–29]. Protocols utilizing pool-split and barcoded oligo ligation strategies greatly improve the throughput of scRNA-seq[30,31]. In this work, we developed single-cell combinatorial hybridization sequencing (CH-seq), another high-throughput and low-cost pool-split-based method, to profile gene expression in single cells. We applied CH-RNA-seq to profile over 1 million cells from 19 tissues in neotenic adult axolotls, metamorphosed adult axolotls and larval stage axolotls. Previous efforts of axolotl tissue-based bulk RNA-seq (TRIzol

extraction) and scRNA-seq identified tissue specific transcripts related to limb regeneration and human diseases[16–18]. Based on these efforts, the adult axolotl cell landscape, created using CH-seq to detect transcripts fixed in dissociated cells, comprehensively covered cell diversity within and between axolotl tissues at single-cell level. The single-cell transcriptomic comparison between neotenic and metamorphosed axolotls allowed us to explore perturbed genes and cell type in metamorphosis. Gene regulatory network (GRN) enabled comparison of potential driven regulators. The whole-organism single-cell transcriptome landscape of the larval-stage axolotl during limb development (Day 30 to Day 70 post fertilization) integrates stepwise dynamics compared with a published limb regeneration dataset. We provide public access to explore the single-cell dataset of the Axolotl Cell Landscape (http://bis.zju.edu.cn/ACA/).

## Results

**Combinatorial hybridization sequencing (CH-seq) is a high-throughput single-cell transcriptome sequencing method**. We developed a combinatorial indexing method named CH-seq (Fig. 1a). CH-seq is based on previously reported pool-split barcoding protocols, including SPLiT-seq[32] and sci-RNA-seq3[31]. Generally, in situ reverse transcription introduces the first round of indices (96 or 384 for barcode #1), oligo hybridization without ligation introduces the second round of indices (768 for barcode #2). After second-strand cDNA synthesis, library amplification generates the last round of indices (i5 indices for barcode #3, i7 indices for barcode #4). Without T4 ligase, total barcode combinations could easily range from 30 million to 0.1 billion depending on the experiment (for a cost of less than $0.01 per cell). Library preparation of 1 million single cells or nuclei can be handled by one researcher in 2 days.

As a proof of principle, we performed a mixed-species CH-RNA-seq experiment on HEK293T (human) and NIH/3T3 (murine) cell lines. We first compared CH-seq NIH/3T3 datasets with ENCODE bulk poly(A) RNA-seq datasets using integrative genomics viewer[33]. Genome read coverage from CH-seq showed a high correlation with published ENCODE data. We compared CH-seq with the other representative scRNA-seq method (10X Genomics). RNA reads in CH-RNA-seq libraries were enriched at upstream regions of transcription termination sites (Fig. 1b). The numbers of unique mapped RNA reads, genes detected in CH-RNA-seq were similar to published pool-split-based methods, including sci-RNA-seq[34] and SPLiT-seq[32] (Fig. 1c). The CH-seq library quality of the human HEK293T and mouse NIH/3T3 mixture displayed a reasonable collision rate and sensitivity (Supplementary Fig. 1f). The reads and gene capture sensitivity of CH-RNA-seq were lower than commercialized 10X genomics platform but higher than commonly used droplet-based method[35].

**Single-cell transcriptome landscape of neotenic and metamorphosed axolotls**. We applied CH-RNA-seq to five natural neotenic adult, and three thyroxine-induced metamorphosed adult axolotls (d/d strain) to create tissue-based single-cell transcriptomic landscape of adult axolotls, and elucidate the cellular and molecular changes occurred in metamorphosis. We collected 19 tissues from one neotenic axolotl or 16 tissues from one metamorphosed axolotl per experiment (Fig. 2a). Libraries in each experiment (one animal) were sequenced in a single channel of a BGI DNBSEQ-T7. After base quality control and removing doublets, a shallow sequencing depth yielded ~3500 reads per cell. On average, we detected 1035 unique molecular identifiers (UMIs) and 254 genes in neotenic axolotl cells and 836 UMIs and 300 genes in metamorphosed axolotl cells (Supplementary

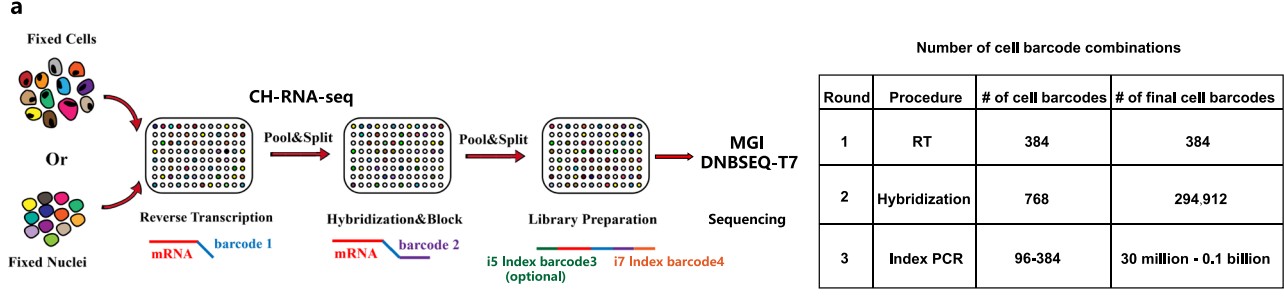

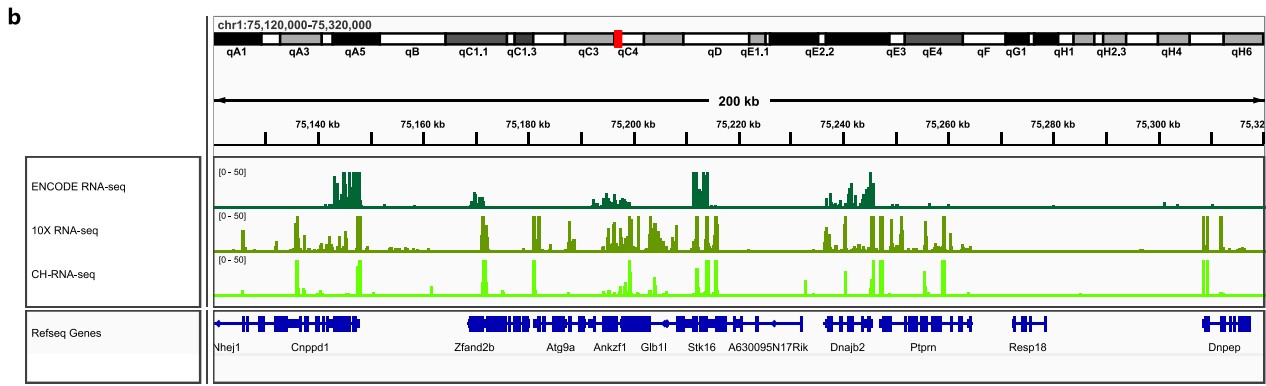

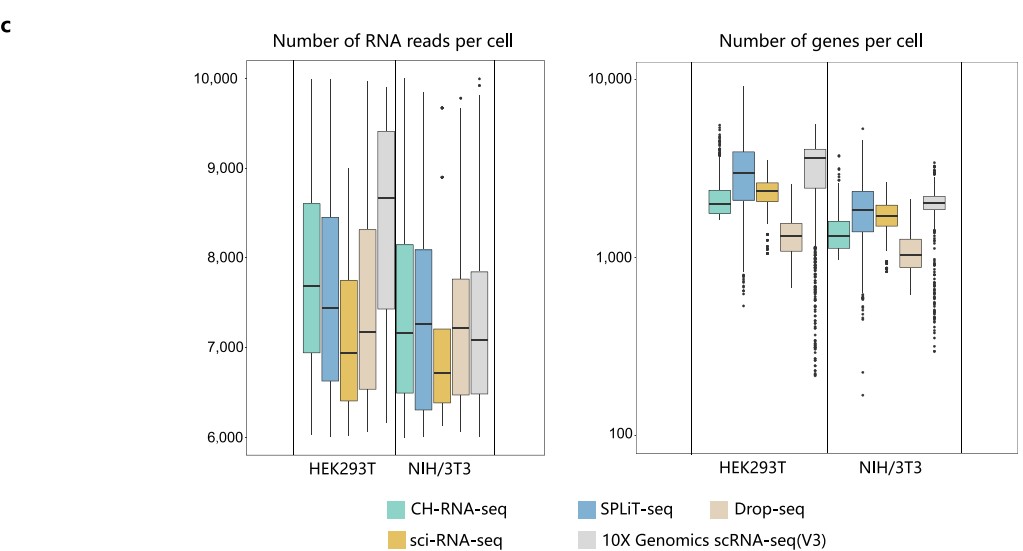

**Fig. 1 Profiling of the transcriptome in single cells using CH-seq. a** Diagram illustrating the experimental workflow for combinatorial hybridization sequencing (RT: Reverse transcription. For a detailed description of the experimental procedure, see Methods). **b** Representative genome browser view of CH-seq NIH/3T3 cell data, 10X Genomics data and ENCODE NIH/3T3 cell data (RNA-seq ("GSE39524") read coverage using an integrative genomics viewer (IGV). ENCODE datasets were obtained from the Gene Expression Omnibus with the accession numbers mentioned above. **c** Box plots showing the number of uniquely mapped RNA reads and the number of genes detected per cell from HEK293T and NIH/3T3 cells ($n = 842$ cells, source data are provided as supplementary data 2, The boxplots are defined by the 25th and 75th percentiles, with the centre as the median, the minima and maxima extend to the largest value until 1.5 of the interquartile range and the smallest value at most 1.5 of interquartile range, respectively.). Original version of the number of genes and reads in sci-RNA-seq ("GSE98561"), SPLiT-seq ("GSE110823"), Drop-seq ("GSE63269") and "10X Genomics scRNA-seq [https://support.10xgenomics.com/single-cell-gene-expression/datasets/3.0.2/1k_hgmm_v3_nextgem]" were obtained as source data in Paired-seq[28] (Supplementary Data 2). The original dataset could be obtained from the Gene Expression Omnibus with the accession numbers mentioned above.

Fig. 1d, e). Altogether, the gene expression of 716,199 adult neotenic axolotl cells and 138,447 adult metamorphosed axolotl cells (UMIs > 200) covered the main cell types in diverse systems of the animal (Fig. 2b, c). Approximately 20% of cells in the first round of experiment were recovered after pool-split process and passed filtration steps. Louvain clustering and t-SNE visualization were applied to profile the single-cell transcriptome landscape

(Fig. 2d, e) as well as major cell types (Fig. 2f, g). We detected quite a number of differentially expressed genes in each cluster of merged datasets with a shallow sequencing depth for downstream analysis (Supplementary Fig. 2a). Cells from experimental replicates showed batch distribution of each organ (Supplementary Fig. 1c). We performed MetaNeighbor[36] analysis of the transcriptome datasets of neotenic axolotls and metamorphosed

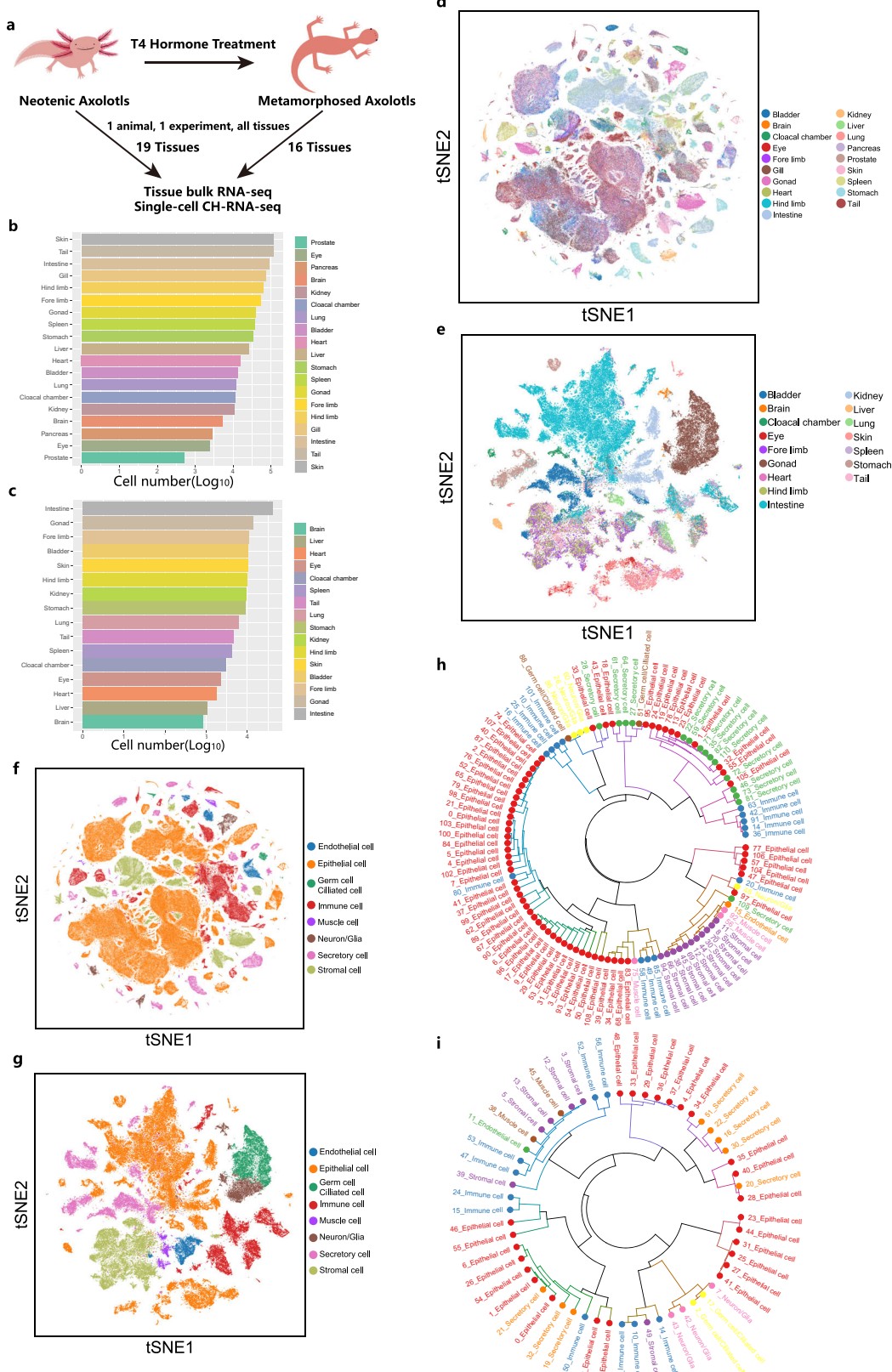

axolotls. The results showed a lineage-specific distribution of major cell types, such as neurons, epithelial cells, secretory cells, endothelial cells, immune cells, and stromal cells (Fig. 2h, i).

To evaluate the overall gene expression patterns in CH-RNA-seq, we compared single-cell RNA-seq data of each organ with a bulk RNA-seq dataset generated by the standard TRIzol protocol.

Reassuring correlations of mean expression were observed across all the tissues in neotenic and metamorphosed axolotls (Supplementary Fig. 1b). Considering morphometric change in metamorphosed axolotls may result from cell death and differentiation in organs, we adopted single-cell lineage inference using cell expression similarity and entropy (SLICE)[37] to predict the cell

**Fig. 2 Mapping the axolotl cell landscape. a** Experimental design of axolotl metamorphosis induction and adult axolotl cell landscape construction (number of biological replicates: neotenic axolotl CH-RNA-seq, $n = 5$; metamorphosed axolotl CH-RNA-seq, $n = 3$; bulk RNA-seq of neotenic axolotls, $n = 2$; bulk RNA-seq of metamorphosed axolotls, $n = 2$). Bar plots showing the number of cells detected in each tissue from adult neotenic axolotls (**b**) and metamorphosed axolotls (**c**) (log10 scale). t-stochastic neighbor embedding (tSNE) plots showing all the single cells profiled using CH-RNA-seq in adult neotenic axolotls (**d**) and adult metamorphosed axolotls (**e**), colored by tissues. tSNE plots showing all the single cells profiled using CH-RNA-seq in adult neotenic axolotls (**f**) and metamorphosed axolotls (**g**), colored by major cell types. Hierarchical trees showing the relationship between cell types in adult neotenic axolotls (**h**) and metamorphosed axolotls (**i**). Each hierarchy was built by performing hierarchical clustering on the area under the receiver operating characteristic (AUROC) scores acquired from MetaNeighbor analyses. Node color indicates the cell lineage for each cell type. The main branches, corresponding to the taxonomy, are annotated with cell lineages.

---

differentiation state of each organ between neotenic and metamorphosed axolotls. Metamorphosed axolotls used in the experiment were at final post-metamorphosis stage (over 23 days post-T4 induction, as described previously in *Ambystoma velasci* (*A. velasci*)), that has been reported have the greatest number of stage-specific transcripts[25]. We found single cells in different tissues exhibited higher entropy in metamorphosed axolotls at post-metamorphosis stage, indicating that cells in metamorphosed axolotl tissues are still undergoing continuous differentiation and may possess higher transcriptional plasticity (Supplementary Fig. 1g).

**Identification of transcriptome-based neotenic and metamorphosed axolotl cell subtypes.** Single-cell transcriptomes of neotenic and metamorphosed axolotl tissues were visualized using uniform manifold approximation and projection (UMAP)[38]. We annotated 459 and 304 subclusters in neotenic and metamorphosed axolotls respectively, based on canonical homologous marker genes[26,39] (Supplementary Fig. 3, 4). We further checked the number of differentially expressed genes in each cluster of all the neotenic and metamorphosed axolotl tissues (Supplementary Fig. 2b, c). Among all the subclusters, we annotated several major cell types across neotenic and metamorphosed axolotl tissues (e.g., immune cells specifically expressing *Ptprc* (*Cd45*); vascular endothelial cells specifically expressing *Cdh5*, *Eng*, and *Lyve1*; stromal cells specifically expressing *Dcn* and *Vim*; epithelial cells specifically expressing *Epcam*).

Previous bulk transcriptome landscape revealed heart and liver enriched ortholog mRNAs among axolotls and humans were related with human diseases[17]. In heart, reported *Tnnt2* and *Myh7* were highly expressed by cardiomyocytes and endothelial cells clusters in our neotenic axolotl heart datasets. Another reported transcription factor *Gata4* was also highly expressed in these clusters as the mutation of *Gata4* is correlated with congenital heart disorder. In liver, previous reported *Lgals2* and *Hpn* were widely expressed in our hepatocyte's clusters. Moderate expression levels of *Serpind1* which correlated with chronic liver diseases in previous study match the restricted expression pattern of *Serpind1* in a specific cluster of hepatocytes in our liver dataset. Another work mapped tissue-based bulk de novo transcriptome assembly of neotenic axolotl and identified limb regeneration factors[16]. Compared with this study, we detect different types of collagens in limb cartilage. Of note, we found a bone-enriched marker, cathepsin k (*Ctsk*) was also highly enriched in limb-restricted macrophages. Generally, tissue-specific transcripts in these works exhibited high correlation with single-cell gene expression datasets in neotenic axolotl tissues (Supplementary Fig. 1a).

Induced metamorphosis in *Ambystoma* species such as axolotls and *A. velasci* is accompanied by morphometric change and tissue remodeling[25,40]. In addition, axolotl metamorphosis brings about a decline in regeneration ability in multiple tissues[8,41]. Along with the change from hydrophilous to terricolous environment, it is intriguing that how targeted perturbation of

cell types correlated with functional changes in axolotl, such as enhancement of limb growth and decline of certain epithelial cells in skin and tail. These cell-type-specific patterns of gene expression in neotenic and metamorphosed axolotls have not been extensively studied. Thus, tissue-level single-cell transcriptomics provides an opportunity to describe the variation between neotenic and metamorphosed axolotls.

Major tissue remodeling events occurred in the respiratory system and motor system[40]. Respiratory system in neotenic axolotl include skin, lung and gill. In the skin, keratinocytes, superficial cell fibroblasts, and basal cells mainly construct the epidermis and dermis. In single-cell RNA-seq datasets, metamorphosed axolotl skin formed defensive secretory cells defined by higher expression levels of *Muc5ac* and *Muc5b*. We experimentally validated differentially expressed genes by RNA in situ hybridization (Fig. 3a, b). The transition of increased *Muc5ac* transcription in mucus-secreting cells was similar to the IL13-induced metaplastic process in human asthma airway epithelial cells[42]. In axolotl lungs, we identified canonical alveolar epithelial cells, ciliated cells and goblet cells as in mammalian lungs. After metamorphosis, one stromal cell cluster in the lung generates a higher proportion of cells that expressed inflammatory regulator *Ptx3*, which plays a protective role in acute lung injury[43] (Fig. 3c; Supplementary Fig. 6b). Connective tissues contain stromal cells that may inhibit cell death and the inflammatory response. We could not detect goblet cells in metamorphosed axolotl lungs, but another type of myoepithelial cell was defined using *Cd109*[44] (Fig. 3c, d). In epithelial cells, *Cd109* could repress the function of TGF-β1 and inhibit epithelial–mesenchymal transition process[45].

In motor system related tissues, the metamorphosed tail demonstrated a similar molecular perturbation pattern to the skin. Tail fin gradually disappeared, and the number of *Krt6a*+ basal epithelial cells greatly increased. Up-regulation of *Krt6a* was also observed in skin basal keratinocytes and forelimb *Col17a*+ epidermal stem cells after metamorphosis (Fig. 3a, e, f). Connective tissue in the limb formed intensively after metamorphosis to support body weight on land. In metamorphosed forelimbs, we could not detect *Muc4*+ mucus-secreting cells. Instead, we defined *Tnmd*+ tenocytes (Fig. 3e, f; see below) and higher proportion of muscle stem cells that uniquely expressed *Pax7*.

In circulation system, we observed multiple cell-type remodeling events in the heart. The neotenic axolotl heart contains *Chga*+ endocrine cells and embryonic cardiac-like cells (*Nfatc1*+, *Cd34*+ and *Postn*+)[46] (Fig. 3g, h; see below). Both neotenic and metamorphosed hearts contain *Gata4*+ endothelial cells that support angiogenesis and are related to cardiomyocyte regeneration[47].

Axolotl tissues in the urinary system, digestive system and nervous system undergo minor remodeling events during metamorphosis. Epithelial layers in the metamorphosed bladder generated urothelial cells that strongly expressed *Akap1* and *Il17b*. In the metamorphosed kidney, we found *Kras*+ epithelial cells

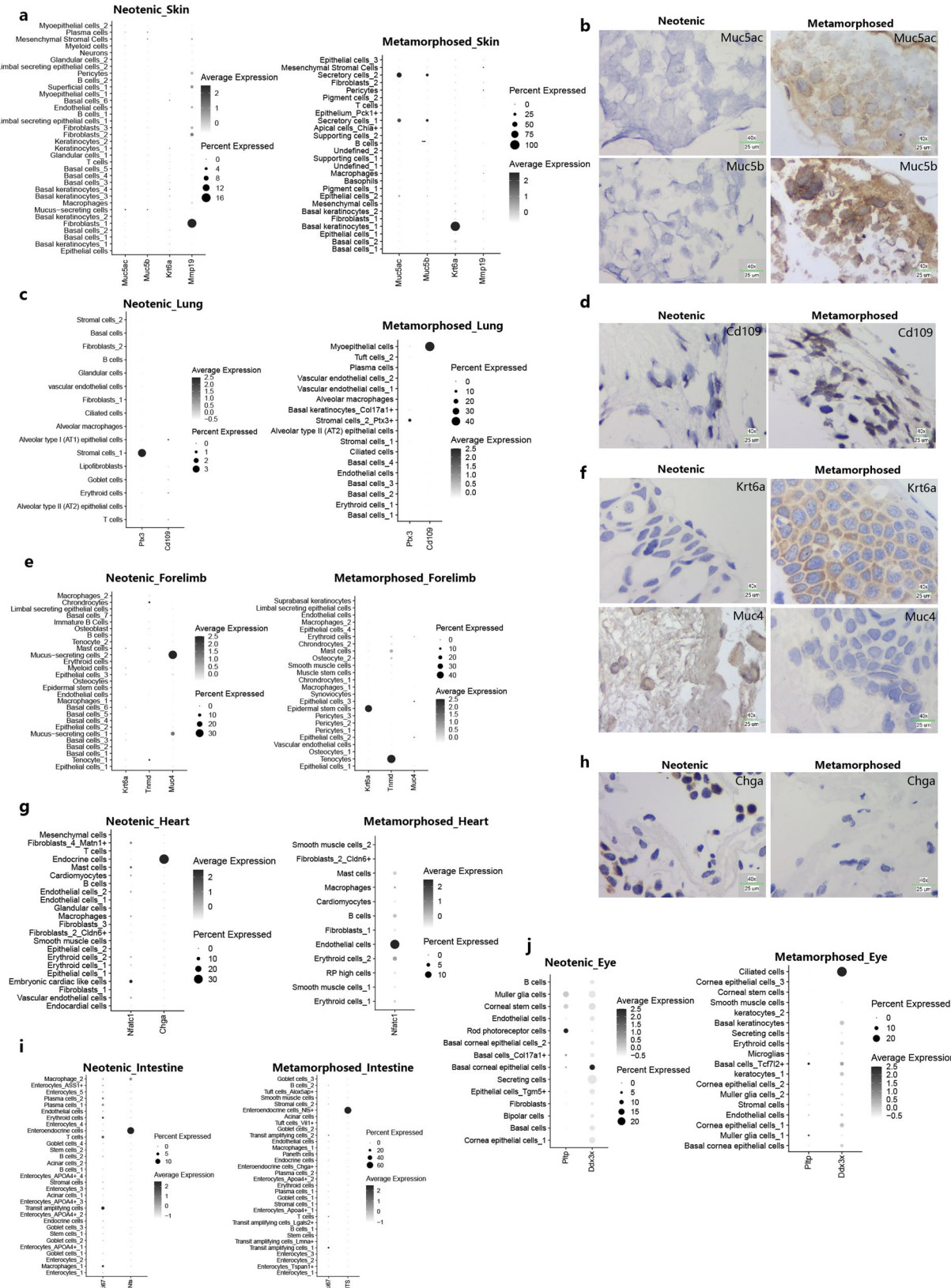

and fibroblasts with unique expressions of *Wfdc5* and *Emid1*. The metamorphosed stomach contains a group of *Gpihbp1*+ capillary endothelial cells (Supplementary Fig. 3, 4). We identified a higher proportion of *Mki67*+ transit amplifying cells in the metamorphosed intestine, and a higher proportion of enteroendocrine cells specifically expressed *Nts*[48] (Fig. 3i).

The thickness and cell size of non-immune cells change during metamorphosis in axolotl eyes[49]. In neotenic axolotl eyes, we identified a neuron subset with specific expression of phospholipid transfer protein (*Pltp*) (Fig. 3j). *Pltp* plays an important role in regulation of tear fluid secretion, and deficiency of *Pltp* results in dry eye syndrome on the ocular surface and damage to the

**Fig. 3 Visualizing of cell clusters marker genes between neotenic and metamorphosed axolotls in major metamorphosed tissues. a, b** Dotplots visualizing expression of genes and representative RNA in situ hybridizations in skin probing for *Muc5ac* and *Muc5b* (Representative images in neotenic metamorphosed axolotls are chosen from two independently animal experiment, scale bars are 25 μm, blue: nuclei). **c, d** Dotplots visualizing expression of genes and representative RNA in situ hybridizations in lung probing for *Cd109* (Representative images in neotenic metamorphosed axolotls are chosen from two independently animal experiment, scale bars are 25 μm, blue: nuclei). **e, f** Dotplots visualizing expression of genes and representative RNA in situ hybridizations in fore limbs probing for *Krt6a*, *Muc4* (Representative images in neotenic metamorphosed axolotls are chosen from two independently animal experiment, scale bars are 25 μm, blue: nuclei). **g, h** Dotplots visualizing expression of genes and representative RNA in situ hybridizations in heart probing for *Chga* (Representative images in neotenic metamorphosed axolotls are chosen from two independently animal experiment, scale bars are 25 μm, blue: nuclei). **i** Dotplots visualizing expression of representative genes in intestine. **j** Dotplots visualizing expression of representative genes in eye.

corneal epithelium[50]. Another cluster of ciliated cells with higher expression level of *Fam188b* and *Ddx3x* was marked in the metamorphosed eye, while *Ddx3x* could inhibit miRNA processing in perturbed photoreceptor development after metamorphosis[51] (Fig. 3j). Overall, these gene perturbation expression patterns of certain cell types support the living environment transition of metamorphosed axolotls.

**Differential genes expression analysis of axolotl tissues after metamorphosis.** We then performed differential genes expression comparisons of major remodeled tissues between neotenic and metamorphosed axolotls. Top upregulated genes in neotenic (avg_$\log_2$FC > 0) and metamorphosed axolotls (avg_$\log_2$FC < 0) were enriched for further gene ontology (GO) analysis (Fig. 4a, b). Uncharacterized axolotl genes without putative orthologous gene annotations were labeled using original gene names.

In the metamorphosed axolotl brain, we observed enriched gene functions of neuron generation and synaptic signaling. Top upregulated genes in the metamorphosed axolotl brain included *Dpp10*, *Rab26* and *Ntsr1* are associated with synaptic vesicles transmission. Differentially expressed genes and enriched biological processes in the eye during metamorphosis are involved in epithelium proliferation and differentiation such as ATF6-mediated unfolded protein response. We observed downregulation of *Rhcg* (associated with pH regulation in aquatic animals) in eye after metamorphosis[25]. Those results suggested a continuous development process of brain neurons and eye epithelial cells in metamorphosis. In metamorphosed axolotl forelimb and skin, we found obvious up-regulation of collagens (*Col5a1*, *Col11a1*, *Col3a1*, *Col6a3*) and specific keratins including *Krt6a*. Representative GO terms are extracellular structure organization and supramolecular fiber organization. Interestingly, fibrinogen encoded by *Fga*, *Fgg*, and *Fgb* which represented blood coagulation, fibrin clot formation was enriched in metamorphosed axolotl liver. Fibrinogen results in the activation of Wnt/β-catenin pathway and enhances the signaling in liver microenvironment[52]. Enhanced liver function in metamorphosed axolotl via up-regulation of fibrinogen may promote fibroblasts, endothelial cells differentiation and migration during other tissues remodeling. In metamorphosed axolotl lung, we identified upregulated gene clusters related to keratinocyte differentiation and epidermal cell differentiation. In the final metamorphosis stage of *A. velasci*, upregulated orthologous genes including *Angptl3* also indicated angiogenesis and alveolarization processes[25]. We performed gene function enrichment of part upregulated genes in metamorphosed *A. velasci* (Supplementary Fig. 5f, g). Top enriched terms suggest ribosome biogenesis processes related to epithelial cells differentiation during lung remodeling. In the metamorphosed heart of axolotl and *A. velasci*, *Gata4* was identified as an important regulator in the differentiation of cardiomyocytes. Function enrichment of top upregulated genes in metamorphosed *A. velasci* heart include striated muscle cell differentiation and positive regulation of lipid biosynthetic process (Supplementary Fig. 5f, g). In the stomach and intestine,

multiple biosynthetic and metabolic processes are enriched before metamorphosis. We found up-regulation of unsaturated fatty acid biosynthetic process and prostaglandin biosynthetic process in metamorphosed axolotl stomach. Endogenous prostaglandins in gastric mucosa could modulate acid secretion as well as mucus and bicarbonate secretion[53]. Prostaglandins secretion may serve as a defensive role during stomach remodeling.

**Tissue specific heterogeneity of nonimmune parenchymal cells.** Next, we studied the inner heterogeneity of nonimmune parenchymal cells across different tissues from neotenic and metamorphosed axolotls. Parenchymal cells, including epithelial cells, secretory epithelial cells, and stromal cells, occupied the majority of cells in scRNA-seq datasets (Fig. 5a, b). In metamorphosed axolotls, higher proportion of stromal cells and muscle cells was observed in tail, skin and limbs. Higher proportion of epithelial cells was observed in lung after metamorphosis. Heatmap of the top markers in each parenchymal cell type revealed distinct expression signatures. Most of these top markers were restricted to one tissue, whereas other markers were conserved across different tissues. In epithelial cells from neotenic axolotls, *Ins* and *Gcg* were merely expressed by the pancreas, and *Ang2* and *Nme7* were enriched in the kidney (Fig. 5e). Other ubiquitously expressed gene modules shared by epithelial cells in different tissues were clustered (Supplementary Fig. 5a). Epithelial cells in many metamorphosed axolotl tissues showed unique expression of *Krt6a*, which is an activated keratin in wound healing[54]. The expression of *Krt6a* was restricted in tail, skin and limbs. In metamorphosed forelimbs, *Krt6a* is expressed in epidermal stem cells defined by *Col17a1*[55] (Fig. 5c). *Col17a1*+ cells were restricted to the skin, tail and eye (Supplementary Fig. 5a). Enriched GO terms in metamorphosed axolotl epithelial cells included hormone level regulation and apoptotic cell clearance (Fig. 5f; Supplementary Fig. 6c).

In neotenic axolotl tissues secretory cells (secretory epithelium), *Acrv1*, *Krt4*, *Krt5*, and *Krt12* were ubiquitously expressed in gills, skin, tails, and limbs (Fig. 5e). Interestingly, *Chga* was uniquely expressed in neotenic axolotl heart endocrine cells (Fig. 3g, h; Fig. 5d). Downregulation of *Chga* in metamorphosed axolotl hearts could increase heart rate and blood pressure and help metamorphosed axolotls adapt to terrestrial life environments[56]. Some neotenic axolotl organs demonstrated high expression levels of ribosomal proteins (Supplementary Fig. 5b). Of note, *Gas6* showed restricted expression pattern in metamorphosed axolotls spleen (ciliated cells) and gonad (Leydig cells), which sets involved in apoptotic cells and macrophage phagocytosis[57,58] (Supplementary Fig. 6a). Differentially expressed gene functions in metamorphosed axolotl secretory cells included transepithelial transport and response to corticosteroids, while neotenic axolotl secretory cells were enriched for neutrophil-mediated immunity (Fig. 5g; Supplementary Fig. 6d).

In neotenic axolotl stromal cells, *Matn1* is differentially expressed in the eye and may be involved in vitreous structure (Fig. 5e)[59]. The expression of *Postn* and *Ptx3* was restricted in

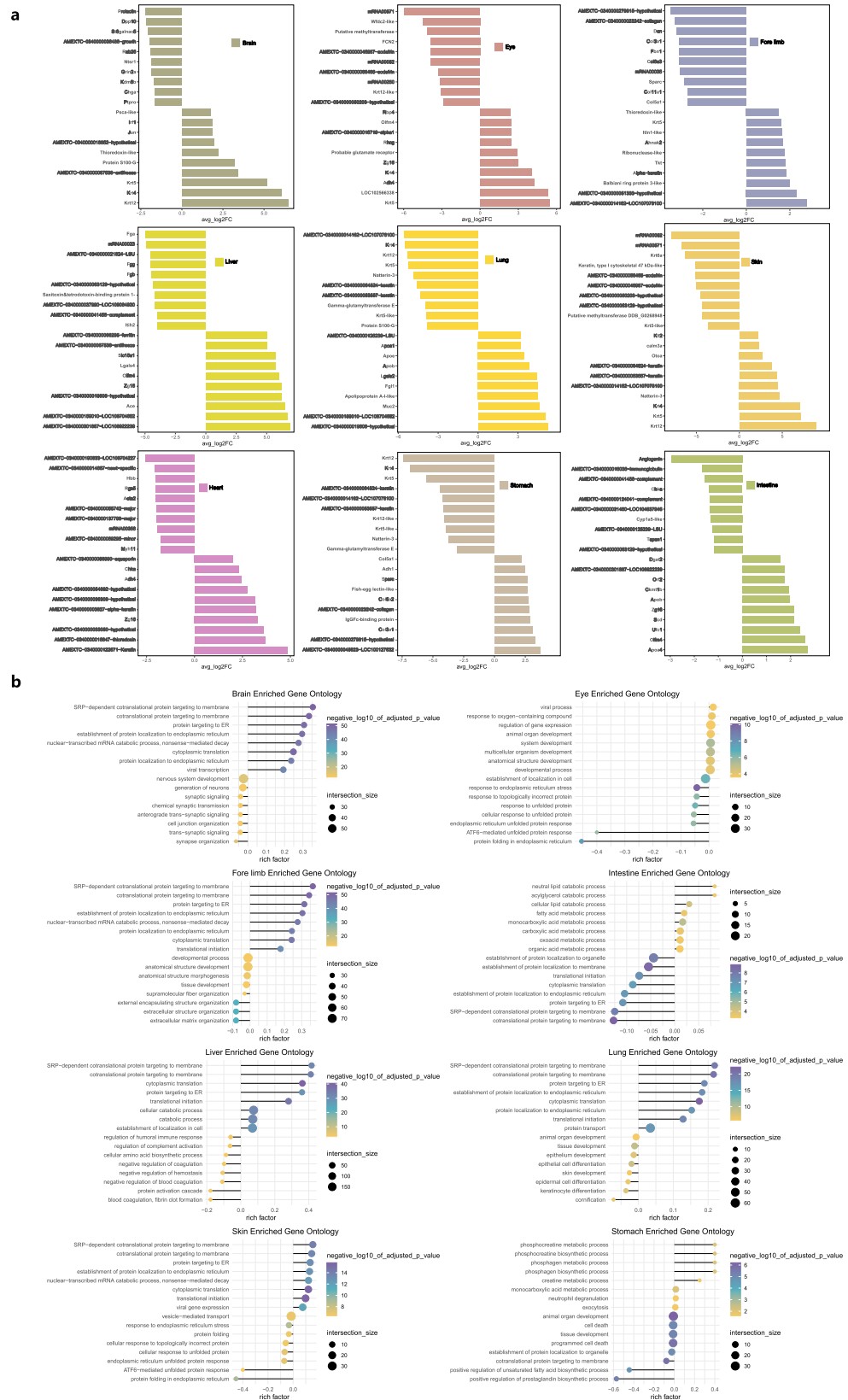

limb, brain, skin and tail fibroblasts (Fig. 5e; Supplementary Fig. 5d). In stromal cells from metamorphosed axolotls, *Lgals2* showed restricted expression in the intestine and cloacal chamber. *Lgals2* plays a role in lipid raft domain stabilization and inflammation suppression[60]. GO terms of metamorphosed axolotl stromal cells enriched genes were ossification, connective

tissue development and muscle organ development (Fig. 5h; Supplementary Fig. 6e). Endothelial cells and muscle cells exhibited a more conserved gene expression module in different tissues (Supplementary Fig. 5e, c). Enriched Gene Ontology biological processes terms of neotenic axolotl endothelial cells included gland development, while transcriptional programs of

**Fig. 4 Differentially expressed genes and their function enrichment between neotenic and metamorphosed axolotls tissues. a** Top 10 differentially expressed genes in major tissues (avg_log2FC: log fold-change of the average expression between neotenic axolotls and metamorphosed axolotls, avg_log2FC > 0: upregulated genes in neotenic axolotls, avg_log2FC < 0: upregulated genes in metamorphosed axolotls). **b** Gene ontology enrichment of differentially expressed genes in **a** (negative_log10_of_adjusted_$p$_value: −log 10 scale of $p$ values, rich factor > 0: function enrichment of upregulated genes in neotenic axolotls with avg_log2FC > 0, rich factor < 0: function enrichment of upregulated genes in metamorphosed axolotls with avg_log2FC < 0, $p$ values were calculated by the hypergeometric distribution, statistical test is one-sided, adjustments $p$ values were made after $p$ value is corrected by Benjamin & Hochberg multiple test).

metamorphosed endothelial cells covered wounding response and actin cytoskeleton organization (Fig. 5i; Supplementary Fig. 6f).

**Cell type perturbation in metamorphosed axolotl skin**. Respiratory system tissues in axolotl including skin, gill, and lung undergo remarkable changes after metamorphosis. The most noticeable differences besides degeneration of gills were observed in skin. The interstitial layer of leydig cells was replaced by cornified squamous epithelial cells originating from basal cells[40]. To study typical thyroid hormone-induced cell-type shifts and perturbations, we adopted the reported "L1 distance" to determine the gene perturbation of newly generated cell types in metamorphosed skin[61]. A previous microarray-based study identified a series of downregulated keratins (*Krt12*, *Krt15*) and upregulated keratins (*Krt4*, *Krt14*, *Krt24*) in skin after axolotl metamorphosis[62]. We downsampled single cells in neotenic axolotl skin and merged single cells in metamorphosed axolotl skin. Reclustering step of those single-cells removed batch effects (Fig. 6a). Epithelial cells in neotenic axolotl skin ubiquitously expressed the basal epithelium markers *Krt4* and *Krt5*, while two clusters of corneal-like differentiated epithelial cells were identified using *Krt12*, *Krt124*, and *Acrv1*[63] (Fig. 6c). *Krt6a* was upregulated in basal keratinocytes (Fig. 6b, d). *Krt6a* showed exclusive spatial distribution in the apical cell layer and was involved in the construction of intermediate filaments[24]. Number of *Mmp19*+ fibroblasts in neotenic axolotls decreased obviously after metamorphosis, suggesting a proliferative disorder-like state of the epidermis[64] (Fig. 6c, d). *Mmp19*-negative fibroblasts with strong proliferation and migration ability may be involved in remodeling of the interstitial layer[65].

We calculated the sum of the absolute differences between centroid coordinates in dimension reduction space (L1 distance). The L1 distance between neotenic and metamorphosed cells within cell types showed an overlapping distribution with the L1 distance between cell types (Fig. 6f), indicating that new cell types may generated in metamorphosis. The largest perturbation distance was observed between fibroblasts (Cluster 10) and the myoepithelium (Cluster 22). We characterized the perturbed gene response in each cell type. Matrix of differentially expressed genes and the cells generated perturbed gene modules (Fig. 6e). Gene ontology functional enrichment of perturbed genes showed a full view of biological processes in skin cell clusters after metamorphosis (Fig. 6g). Fibroblasts (Cluster 10) and myoepithelial cells (Cluster 22) broadly shared perturbed genes in modules C1, C3, C4, and C9 related to signal transduction by p53 class mediators, DNA duplex unwinding and oxidative demethylation in cell differentiation. The distribution of perturbed genes across cell types showed that most perturbed genes were cell type specific (Fig. 6h). Fibroblasts (Cluster 10) enriched the highest number of perturbed genes across all modules, while myoepithelial cells (Cluster 22) enriched many fewer perturbed genes, suggesting that the scale of perturbed genes in cell types was not the only potential regulation paradigm in metamorphosis.

**Gene regulatory network (GRN) construction in neotenic and metamorphosed axolotls**. To understand the regulatory factors that control the differentially expressed genes altered in metamorphosed axolotls. We introduced bigSCale[66] to infer the GRN of neotenic and metamorphosed axolotls (Fig. 7a). The GRN topology of the metamorphosed axolotl harbored relatively many higher nodes, edges and densities. Strong gene–gene relationships within cell lineages in stromal and epithelial cells were observed in the metamorphosed axolotl GRN, while the neotenic axolotl GRN exhibited a fairly high degree of cross-lineage communication. In the metamorphosed axolotl GRN, major lineage communications were driven by skeletal muscle cells and ciliated epithelial cells. In neotenic axolotls, one major network of cross-lineage communication was driven by vascular endothelial cells, fibroblasts, osteoblasts, and smooth muscle cells, and the other was driven by specialized epithelial cells such as glandular cells and enterocytes. We used PageRank centrality to assess the effect of genes on GRN topology and their biological regulation function. We identified the top 50 regulatory genes in the neotenic axolotl GRN and the top 100 regulatory genes in the metamorphosed axolotl GRN ranked by PageRank (Fig. 7b). In the neotenic axolotl GRN, these genes were mainly transcription factors and receptors, including transcription activators of specific cell lineages, such as *Gata4*, *Gata6* and *Pdgfra*. In the metamorphosed axolotl GRN, enriched genes with high centrality include the Wnt signaling activator *Caprin2* and the environmental stress activator *Ppm1d*[67]. Gene ontology enrichment of the top transcription factors in the neotenic axolotl GRN included epithelial-to-mesenchymal transition involved in endocardial cushion formation, gland development, sensory organ development and wound healing (Fig. 7c). Enriched biological process terms in the metamorphosed axolotl GRN included collagen fibril organization, muscle structure development, bone development and tissue morphogenesis (Fig. 7c). These findings highlight the altered transcriptional regulation in metamorphosed axolotls during tissue remodeling and identify potential lineage specific regulators (*Caprin2* and *Ppm1d*) with significant centrality rather than expression levels.

**Limb bud development in larval-stage axolotls**. Prior research has identified quantitative proteomic differences underlying the differential limb regenerative capacity of neotenic and metamorphosed axolotls[22]. However, limb bud development in neotenic axolotls has not been well studied. The forelimb buds are slightly outlined at around Day 8 post-fertilization. The hindlimb buds emerged at Day 30 post-fertilization. We performed whole-organism CH-RNA-seq to profile 217,781 single-cell transcriptomes of neotenic axolotls at 5 time points spanning over 40 days during limb development (Day 30, Day 35, Day 45, Day 50 and Day 70 post-fertilization) (Fig. 8a; Supplementary Fig. 7d). To generate a time-resolved transcriptional landscape, we explored cell lineage visualization using SPRING[68] (Fig. 8b; Supplementary Fig. 7a). Annotation of cell types using specific marker genes identified major lineages of endothelial cells (*Cd34*+, *Cdh5*+), epithelial cells (*Epcam*+, *Krt4*+), stromal cells (*Vim*+, *Dcn*+), muscle cells (*Tagln*+, *Acta2*+), hepatocytes (*Apob*+, *Apoa1*+), astrocytes (*Gfap*+, *Fabp7*+), and germ cells (*Tekt2*+, *Tekt4*+). In addition, we identified limb development

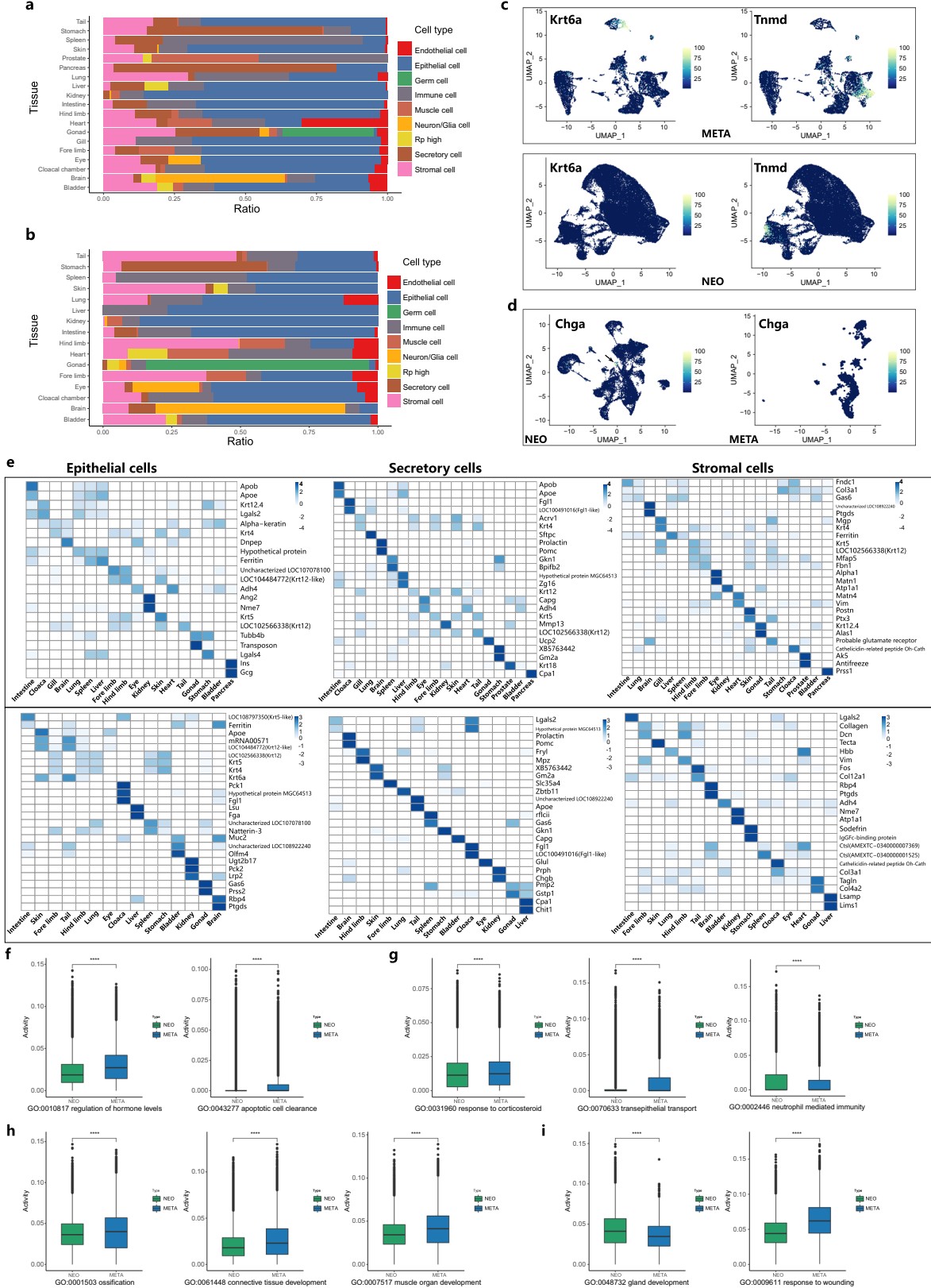

related cell lineages, including myotubule-forming cells (*Sox8*+, *Cirbp*+, *Lama4*+), osteoblasts (*Matn2*+, *Postn*+), chondrocytes (*Sox9*+, *Matn1*+), tendon cells (*Thbs4*+, *Nucb2*+) and articular chondrocytes (*Phospho1*+, *Itga10*+). In bulk level RNA-seq, *Cirbp* has been identified as an anti-apoptotic protein enriched in blastema at early stage after limb amputation[16].

We reclustered chondrocytes, osteoblasts, skeletal muscle cells and tendon cells in our dataset and cells in blastema from 18 to 38 days post amputation respectively[18] (Fig. 8c). MetaNeighbor correlation analysis of limb bud cells and blastema cells was performed[36] (Fig. 8d). Major cell lineages, including chondrocytes, osteoblasts and skeletal muscle cells, showed a high

**Fig. 5 Inner heterogeneity of major nonimmune parenchymal cell types in neotenic and metamorphosed axolotls.** Bar plots showing the fraction of cells per tissue derived from annotated major cell types in neotenic axolotls (**a**) and metamorphosed axolotls (**b**). **c** Feature plots visualization of *Krt6a* and *Tnmd* in neotenic axolotl (NEO) and metamorphosed axolotl (META) forelimbs. **d** Feature plots visualization of *Chga* in neotenic axolotl (NEO) and metamorphosed axolotl (META) hearts. **e** Heatmaps showing the top differentially expressed marker genes across tissues in three major cell types from adult neotenic axolotls (top: NEO) and metamorphosed axolotls (bottom: META). The color represents the expression level. Bar plots showing the activity of selected gene ontology (GO) enrichment terms in epithelial cells (**f**) (META: $n = 54,036$ cells, NEO: $n = 458,162$ cells), secretory cells (**g**) (META: $n = 24,711$ cells, NEO: $n = 73,786$ cells), stromal cells (**h**) (META: $n = 27,466$ cells, NEO: $n = 77,572$ cells), endothelial cells (**i**) (META: $n = 3267$ cells, NEO: $n = 23,924$ cells) from neotenic axolotls (NEO, green) and metamorphosed axolotls (META, blue) ($n =$ number of cells, the boxplots are defined by the 25th and 75th percentiles, with the centre as the median, the minima and maxima extend to the largest value until 1.5 of the interquartile range and the smallest value at most 1.5 of interquartile range, respectively. "****": $p$ values < 0.001, in all cases, $p$ value < 2.22e-10, $t$ test was introduced, adjustments $p$ values were made after $p$ value is corrected by Benjamin & Hochberg multiple test).

correlation, indicating the similarity of transcription programs between limb development and regeneration. Cell lineages in blastema demonstrate higher randomness than developmental limb buds. We then investigated the dynamic gene expression patterns at five time points. Upregulated and downregulated genes consistent with developmental time were examined (Fig. 8e, f; Supplementary Fig. 7b, c). Enriched gene module functions at an early time point (Day 30 post-fertilization) were associated with the development of central nervous system development, epithelial differentiation and myoblast fusion (Fig. 8g). *Wfdc5* plays a critical role in the skin innate immune response and host defense[69]. *Crybb3*, *Crb2* and zinc efflux transporter *Tmem163* are located in the eye and brain. Decreased expression of *Znf536* may enhance neuronal differentiation[70]. In contrast, gene function enrichment at the latter time point included chondrocyte development, osteoblast proliferation and collagen organization (Fig. 8h). *Matn1* and *Csgalnact1* participate in cartilage formation in axolotl larval limb development[71,72]. Interestingly, we observed consecutive upregulation of *A2m*. The transcriptional level of *A2m* decreased in human blood with age and could suppress central signaling pathways in tumor development[73]. Highly conserved *A2m* also inhibits tumor growth in another neotenic animal, the naked mole-rat[74]. The similar antitumorigenic function of *A2m* in axolotls could be a potential novel feature of neotenic animals.

## Discussion

Despite their importance in decades of development and regeneration studies, the molecular basis of axolotls has not been well studied. The whole genome assembly of axolotl and a chromosome-scale assembly of the axolotl genome in recent years have made axolotl a potential model species[14,15]. Bryant et al. created high quality multi-organ axolotl transcriptome assembly and identified key regulators in limb regeneration including *Cirbp* and *Kazald1*[16]. Caballero-Perez et al. presented another diverse transcriptomic datasets of multiple axolotl organs[17]. They analyzed tissue-specific protein coding mRNAs and their potential function in human disease. In addition, they explored conserved microRNAs and putative novel microRNAs in different tissues. Advances in single-cell technology also pushed forward the study of axolotl limb regeneration[18]. Our single-cell transcriptome landscape of larval axolotl limb development constitutes another valuable resource to characterize the cell types and dynamic gene expression patterns in the field. For example, the cell composition of blastema after limb amputation demonstrated high similarity with development limb bub cells, including different chondrocytes, skeletal muscle and tendon cells. Compared with other species, upregulation of conserved gene features during limb development, such as *A2m* and *Matn1*, may correlate with the neotenic development state in axolotls and naked mole-rats[74]. Dedifferentiation of fibroblasts activated developmental programs of limbs in axolotls but not in African clawed frogs[23].

Regeneration competence in African clawed frogs is correlated with secreted inhibitory factors[75]. One limitation of our study is the lack of data on embryonic development to map the global view of neotenic axolotl organogenesis. However, all current evidence suggest a potential correlation between the neoteny fate of axolotls and their regeneration ability. Unlike some amphibians with spontaneous metamorphosis, neotenic axolotl goes through a metamorphosed state under specific natural conditions or by artificial induction using thyroid hormones. Experimentally induced metamorphosis in axolotls could somehow reduce their regeneration ability and lifespan[8,10]. Previous transcriptional study of metamorphosed axolotls and its close relative *A. velasci*, which naturally undergo metamorphosis, compared transcriptional programs in some tissues[24,25]. Here, we constructed a tissue-based single-cell transcriptome landscape of neotenic and metamorphosed axolotls. The adult axolotl cell transcriptomic landscape is the first to demonstrate the comprehensive cellular composition between neotenic and metamorphosed axolotls at single-cell level. We also compared differential gene expression patterns in major parenchymal cell types across all tissues (Supplementary Data 5). We identified specific cell-type shift events after varying degrees of tissue remodeling (Fig. 3). Enrichment of squamous epithelial cells and mucus-secreting epithelial cell type shifts in the tail, skin and limbs support the adaptation of the terrestrial life environment in metamorphosed axolotls.

Gene regulatory networks between neotenic and metamorphosed axolotls further discovered the driven gene modules in tissue remodeling beyond the gene expression level (Fig. 7; Supplementary Data 6). We identified transcription regulation genes in major cell types. In neotenic axolotls, these regulators maintain a stable cell lineage fate, whereas target gene transcription in metamorphosis switches the cell fate regulatory network to another model. Such a switch in transcription regulation may triggers the acceleration of aging and a decline in regeneration ability. One aim to study regeneration in neotenic axolotls is to discover the connection between regeneration, aging and cancer[5,76]. Neoteny in naked mole-rats also results in longevity and cancer resistance rather than regeneration ability[4,77]. Thus, it is of great importance to evaluate the metabolic features and molecular features associated to neoteny in axolotls, in order to clarify the correlation between neotenic and metamorphosed animals and their influence in regeneration and aging in future studies.

In addition to the limitations mentioned above, our study generated relatively sparse transcriptome data with shallow sequencing. This could result in the absence of rare cell types, especially in tissues with a small number of cells. Second, many axolotl tissues have not previously been studied at the single-cell level. Cell type annotation with homologous published marker genes may lead to inaccurate results. We will provide a more comprehensive annotation of axolotl cell types based on feedback from online websites and further multi-omics studies.

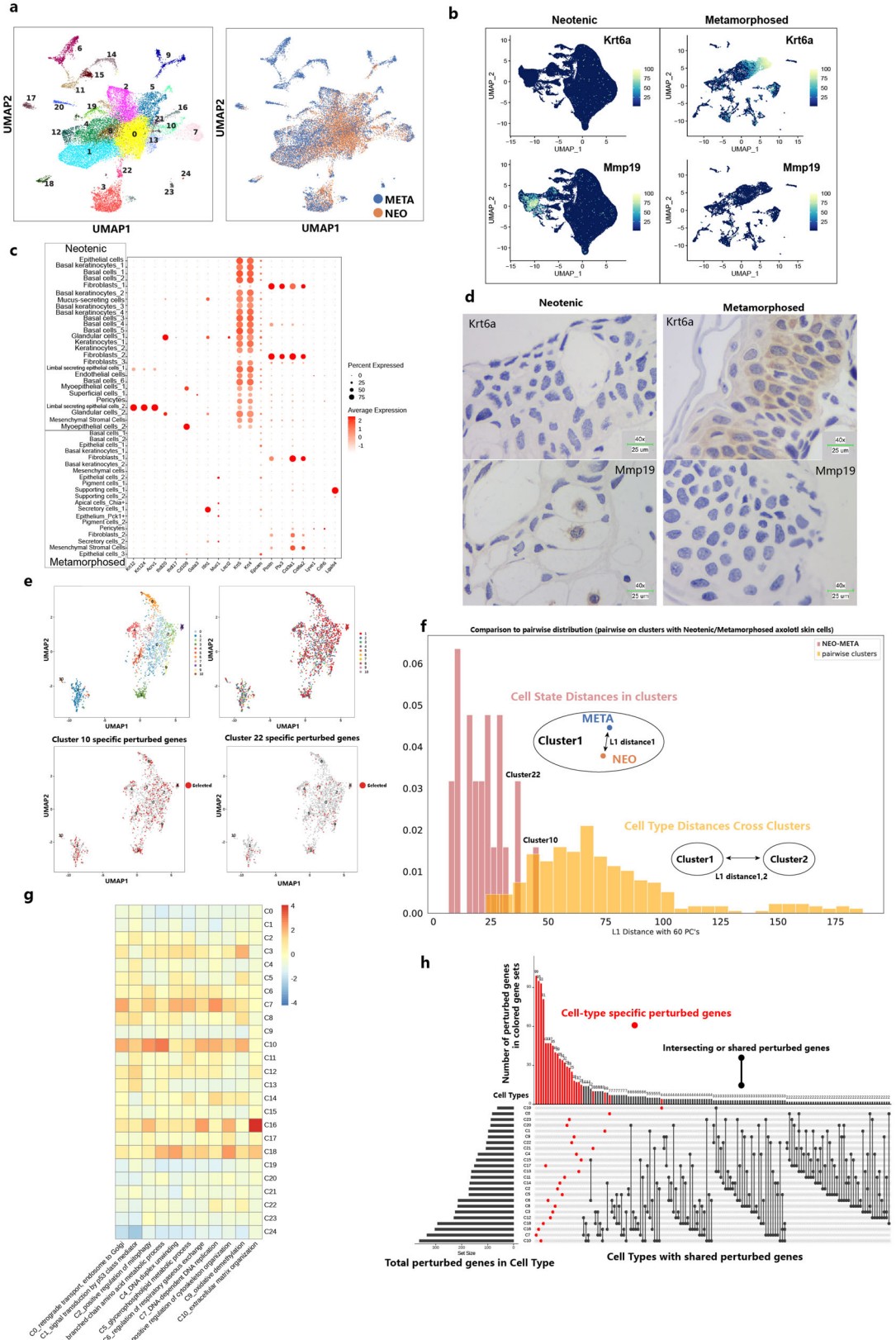

In conclusion, we introduced CH-seq as a new cost-effective protocol for high-throughput single-cell sequencing. We revealed novel insights into neotenic and metamorphosed axolotls. We profiled the transcriptomes of over 1 million cells from adult and larval axolotls spanning Day 30 to Day 70 post-fertilization. The resulting adult axolotl cell landscape covered nearly 500 subtypes of cells across 19 tissues as well as their gene regulatory network. Based on previous tissue-based transcriptome studies on neotenic axolotl and metamorphosed axolotl, we further revealed cell-type perturbation in remodeled tissues. Generally, Secretory epithelial cells in metamorphosed skin, tail and limbs enriched different types of mucins and keratins in order to adapt to the terrestrial

**Fig. 6 Perturbation of skin cell types in response to metamorphosis. a** UMAP visualization of downsampled single cells from neotenic axolotl skin and all single cells from metamorphosed axolotl skin (right: UMAP plot colored by neotenic axolotl (NEO) and metamorphosed axolotl (META); left: UMAP plot showing the cluster numbers). **b** Feature plots visualization of *Umod*, *Krt6a* and *Mmp19* in original neotenic axolotl skin samples (left) and original metamorphosed axolotl skin samples (right). **c** Dotplot showing selected marker gene expression in the original skin cluster (Neotenic: original skin clustering of neotenic axolotl skin in Supplementary Fig. 3; Metamorphosed: original skin clustering of neotenic axolotl skin in Supplementary Fig. 4). **d** Representative RNA in situ hybridizations in skin probing for *Krt6a* and *Mmp19* (blue: nuclei, representative images in neotenic metamorphosed axolotls are chosen from two independently animal experiment, scale bars are 25 μm). **e** UMAP visualization of extracted perturbed differentially expressed genes from merged clustering of single cells (top right: same UMAP plot in which genes were colored by perturbed cell types; bottom: highlight of perturbed genes in Cluster 10 and Cluster 22). **f** Histogram of L1 distances between centroids of neotenic and metamorphosed axolotl cells within each merged cluster versus pairwise L1 distances between centroids of all merged clusters. Cluster 10 and Cluster 22 harbored the largest internal distances and overlapping distances. **g** Gene ontology (GO) enrichment of perturbed genes in each merged cluster. **h** UpSet plot visualization for intersecting sets of "perturbed" genes (left bar plot: number of differentially expressed perturbed genes under metamorphosis in each merged cluster; The connected dots represent overlaps between perturbed genes in each cluster, differentially expressed perturbed genes in only one cluster are colored in red; top bar plot: number of perturbed genes in red gene module).

environment. Differentiation of muscle cells and mesenchymal cells in limbs supported crawling movement in metamorphosed axolotls. *Nfatc1*+ embryonic cardiac-like cells and mesenchymal cells suggested stronger differentiation potential in neotenic axolotl heart. Up-regulation of fibrinogen in metamorphosed axolotl liver enhances liver function and migration of other cells. In the digestive system, secretion of protective endogenous prostaglandins in the stomach also have a positive effect on gastric mucosa. Increased transit amplifying cells in the intestine indicated more active state of intestine function after remodeling. These foundational datasets serve as important resources for developmental and regenerative biology (http://bis.zju.edu.cn/ACA/). The adult axolotl cell landscape and larval stage axolotl cell landscape lay a foundation for tissue-level molecular studies of axolotls. These resources could facilitate future exploration of neoteny, regeneration, aging and cancer research in axolotls as well as cross-species comparison.

## Methods

**Axolotl strains, husbandry, and induction of metamorphosis.** The d/d strain axolotls were provided by Ji-Feng Fei's laboratory in the Department of Pathology, Guangdong Provincial People's Hospital, Guangdong Academy of Medical Sciences, Guangzhou, China. Axolotls were housed in glass containers at room temperature (20 °C). The anti-chloride-treated Holtfreter's solution was replaced every day.

For metamorphosis induction, we randomly chose adult axolotls (12 months old) and transferred them into new containers. L-Thyroxine (T4, Sangon Biotech, Cat# A602869) was used as described elsewhere. Briefly, 50 nM T4 hormone was provided in Holtfreter's solution, and the medium was renewed every day. After approximately 12 days of treatment with T4, we observed the degeneration of gills, fins and weight loss compared to those remaining in neoteny. Metamorphosis was then maintained for another 2 weeks until the gills completely disappeared. The liquid level was gradually reduced and allowed metamorphosed axolotls to breathe in the air.

For adult samples, all the other tissues were collected from 12-month-old axolotls. Larval samples were collected from axolotls at 30 days post-fertilization to 70 days post-fertilization.

All experiments performed in this study were approved by the Animal Ethics Committee of Zhejiang University (Lot number: ZJU20210135). All experiments conformed to the relevant regulatory standards at Zhejiang University Laboratory Animal Center.

**Tissue dissociation and cell preparation.** For cultured human and mouse cell lines used in mixed-species experiments, 3T3 and 293T cells were cultured in DMEM medium (Gibco) with 10% fetal bovine serum (FBS, Thermo Fisher) and 1% penicillin–streptomycin (Thermo Fisher). Cells were harvested by trypsinization and resuspended in cold 1× PBS.

All animals were anesthetized by 0.1% ethyl 3-aminobenzoate methanesulfonate salt (Sigma-Aldrich) in water prior to heart perfusion with 0.8× PBS. For each experiment, we collected tissues as discussed above from adult axolotls and immediately placed them into 0.8× ice-cold DMEM (Gibco).

For CH-RNA-seq, tissues were then minced into ~1 mm pieces on ice with scissors and transferred to a 15 ml RNase-free centrifuge tube, then centrifuged at 300 g for 5 min at 4 °C. The supernatants were removed, and the pellet was resuspended in 5 ml of dissociation enzyme mix (1 mg/ml enzyme for each tissue,

1 mg/ml DNase I, 5 mM CaCl$_2$, supplied in 0.8× DMEM). Tissue pieces were pipetted up and down gently every 15 min during dissociation at room temperature, and ice-cold 0.8× DMEM was added to stop the dissociation. The medium was then filtered using a 40 μm strainer (Biologix), and the cells were washed twice with 0.8× ice-cold PBS and 0.8× prechilled in PBS (0.8× PBS with 1% murine RNase inhibitor (Vazyme Biotech), 1% BSA (Sangon Biotech), and 0.1 mM DTT (Sangon Biotech)). Pellets were resuspended in 100 μl PBS and 10 ml 4% PFA–0.8×PBS was then added carefully. Cells were fixed on ice for 15 min and washed with 1 ml PBS. After being pelleted by centrifugation at 500 × g for 5 min, cells could then be resuspended with PBS at a density of 2 million per milliliter and stored at −80 °C for at least a week.

**Protocols of CH-RNA-seq.** For CH-RNA-seq, frozen cells of each tissue were thawed at 37 °C and centrifuged at 500 × g for 5 min first. Then, fresh fixed or thawed frozen cells were resuspended with 100 μl PBS, and 400 μl Triton X-100 in PBS (10 μl 10% Triton X-100 in 390 μl PBS) was added. The mixture was incubated on ice for 3 min for permeabilization. Next, the cells were pelleted and resuspended with 500 μl DEPC-treated water, and 3 ml 0.1 N HCl was added slowly. After 5 min of incubation on ice, 3.5 ml Triton X-100 in Tris-HCl (35 μl 10% Triton X-100 in 3.5 ml 1 M Tris-HCl, pH 8.0) was added to quench permeabilization. Cells were then washed once with 1 ml of PBS and filtered with a 40 μm strainer. Cell density was determined using a blood cell counting chamber. For reverse transcription, cell numbers for each tissue were estimated, and the corresponding volume of cells in PBS was split into four 96-well plates. For each well, 6.5 μl cells (from 5000 to 20,000) was mixed with 0.5 μl 10 mM dNTPs (Thermo Fisher Scientific) and 0.5 μl 100 μM RT barcode primer (see Supplementary Data 1). The mixture was incubated at 65 °C for 5 min and immediately placed on ice. Two microliters of 5× RT buffer (31 mM Tris-HCl(pH 8.0), 37.5 mM NaCl, 3.1 mM MgCl$_2$, 10 mM DTT), 0.5 μl Maxima H Minus RTase (Thermo Fisher Scientific) and 0.1 μl Murine RNase inhibitor (Vazyme Biotech) were premixed, and 2.5 μl RT mixture was added to each well. Reverse transcription was carried out by incubating the plates at gradient temperature: 3 cycles (8 °C for 12 s, 15 °C for 45 s, 20 °C for 45 s, 30 °C for 30 s, 42 °C for 2 min, and 50 °C for 3 min) and 50 °C for 60 min. After reverse transcription, plates were placed on ice for 1 min to stop the reaction. All reagents were pooled together into a 15 ml centrifuge tube and then centrifuged at 500 g for 5 min. Pellet was washed with PBS twice and resuspended with hybridization buffer (50 mM Tris-HCl, 10 mM MgCl$_2$, 10 mM DTT, 0.1% Triton X-100, 10% PEG8000 (Sigma-Aldrich), 1% Murine RNase inhibitor). Hybridization buffer within cells was split into eight 96-well plates (3 μl for each well) and 2 μl 25 μM pre-annealing hybridization primers (50 μM HY head oligo, 50 μM barcoded HY primer oligo, mixed equally and incubated at 95 °C for 2 min, then slowly cooled to 25 °C with a temperature ramp of −0.1 °C/s) were added to each well (see in Supplementary Data 1). Plates were incubated at 37 °C for 90 min, and 0.5 μl of 100 μM block tail primer oligo was added to block any redundant hybridization primers. After blocking for 30 min at 37 °C, all reagents were pooled into a 15 ml centrifuge tube and then centrifuged at 500 g for 5 min. The pellet was washed with PBS twice and resuspended in 40 μl of PBS. Then, 60 μl PNK mix (10 μl 10× PNK buffer (NEB), 20 μl T4 polynucleotide kinase (NEB), 10 μl 10 mM ATP (NEB), and 20 μl DEPC-treated water) was added. The PNK reaction was carried out at 37 °C for 30 min. Then, 1 ml ice-cold PBS was added to stop the reaction. Cells were then filtered using a 40 μm strainer. The reagent was centrifuged at 500 g for 5 min after discarding the supernatants. The pellet was again resuspended in PBS, and the density of cells was estimated. After splitting into one or more 96-well plates. There were 5000–8000 cells per well. The volume was adjusted to 8 μl. Second strand synthesis mix (1.33 μl 10× buffer, 0.66 μl second strand synthesis enzyme mix) was added into each well and the plates were incubated at 16 °C for 3 h (stop point at 4 °C). Then, 10 μl cell lysis buffer (20 mM Tris pH 8.0, 400 mM NaCl, 100 mM EDTA, 4.4% SDS (Sangon Biotech)) and 2 μl proteinase K (Sangon Biotech) were added to the wells. Cell lysis was performed at 55 °C for 60 min. Then, 1 μl PMSF (1 mM) was added to quench the lysis reaction. The plates were incubated at 37 °C for

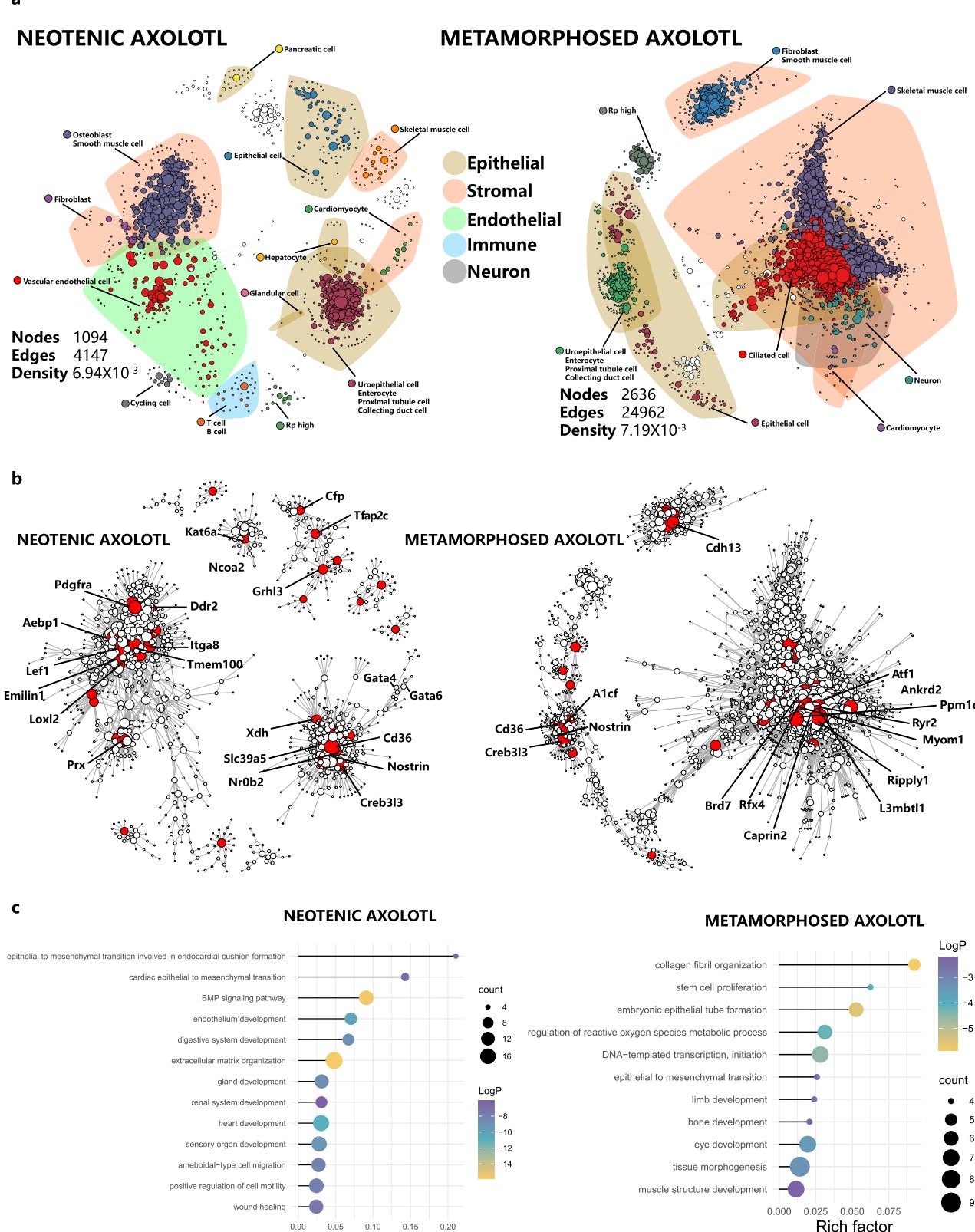

10 min. For each well, the mixture was then purified using 1.5X VAHTS DNA Clean Beads (Vazyme Biotech). Then, 10 μl of product in DEPC-treated water was transferred into new 96-well plates. For each plate, we randomly chose ten wells to quantify the dsDNA concentration using Equalbit3.0. Tn5 transposase from the TruePrep DNA Library Prep Kit V2 (Vazyme Biotech) was used for cDNA tagmentation. Then, 10 μl of product in DEPC-treated water was transferred into a new 96-well plate. For each well, 14 μl PCR mix (12 μl 2X KAPA HiFi HotStart

ReadyMix (Kapa Biosystems), 1 μl 10 mM P5 primer, and 1 μl 10 mM indexed P7 primer, see Supplementary Data 1) was added to the plates. The PCR program was as follows: 72 °C for 5 min; 98 °C for 30 s; 12 cycles of 98 °C for 10 s, 60 °C for 30 s, and 72 °C for 1 min; 72 °C for 5 min; and a 4 °C hold. After pooling PCR products, two rounds of size selection with VAHTS DNA clean beads (Vazyme Biotech) were used to purify the DNA library between 300 and 500 bp. The libraries were finally mixed, and the concentration of dsDNA was quantified.

**Fig. 7 Gene regulatory network analysis of neotenic and metamorphosed axolotls. a** Gene regulatory networks of neotenic axolotls and metamorphosed axolotls. Nodes represent network genes, and edges represent putative relationships. The size of nodes represents PageRank centralities. Calculated clusters are colored, and cell-type annotations are highlighted. Polygon shape covered annotated major cell types. **b** Same gene regulatory networks with the top 50 nodes ranked by PageRank centrality in the neotenic axolotl network and the top 100 nodes ranked by PageRank centrality in the metamorphosed axolotl network. Top nodes were highlighted in red. **c** Selected gene ontology (GO) enrichments of the top 100 differential PageRank centrality ranked nodes in the neotenic axolotl network and metamorphosed axolotl network (p values was calculated by the hypergeometric distribution, statistical test is one-sided, adjustments p values were made after p value is corrected by Benjamin & Hochberg multiple test).

---

**MGI library preparation and sequencing**. The purified linear DNA library was circularized into a single-stranded DNA (ssDNA) library using a VAHTS® Circularization Kit for MGI (Vazyme Biotech). Then, the ssDNA library was amplified using a DNBSEQ DNB preparation kit (MGI). Amplified DNA nanoballs (DNBs) were sequenced with custom TM (Tn5 modified) sequencing primers on MGI DNBSEQ-T7 with a dark reaction model (CH-RNA-seq: 51 cycles of read1 with a dark reaction from 11 to 33,100 cycles of read2).

**Axolotl tissue bulk RNA library preparation and sequencing**. Paired tissues in axolotl single-cell experiments from neotenic axolotls ($n = 2$) and metamorphosed axolotls ($n = 2$) were collected. Total RNA was purified using TRIzol. Following reverse transcription, second strand DNA synthesis, transposase tagmentation and library construction, bulk RNA libraries of each organ were pooled and sequenced on an Illumina HiSeq X Ten in paired-end mode (paired-end 150 bp model).

**RNA in situ hybridizations**. RNA hybridization experiments were performed as described in this protocol (https://www.protocols.io/view/rna-in-situ-hybridization-p33dqqn?step=1)[19]. Treated samples were visualized by confocal microscopy (FV1000, Olympus) ($n = 2$, three times the experiment shown was replicated in the laboratory). Oligo sequences of RNA probes and primers are listed in Supplementary Data 4.

**Processing of the RNA-seq data**. The raw fastq-format sequencing data from a DNBSEQ-T7 were first split into i7 indexed sub-libraries using "splitBarcode [https://github.com/MGI-tech-bioinformatics/splitBarcode]"). For each sub-library, the revised Drop-seq core computational tool was used for "data pre-processing [http://mccarrolllab.org/wp-content/uploads/2016/03/Drop-seqAlignmentCookbookv1.2Jan2016.pdf]"[35]. We used STAR (version 2.5.2a)[78] with default parameters for mapping 3T3 cells and 293T cells. Reads from 3T3 cells and 293T cells were aligned to a merged hg19-mm10 reference genome (provided by Drop-seq group, "GSE63269 [https://www.ncbi.nlm.nih.gov/geo/query/acc.cgi?acc=GSE63269]"). To save mapping time, reads from axolotls were aligned to the "axolotl genome [https://www.axolotl-omics.org/assemblies]" (V3.0.0) using another mapper bowtie2[79]. GTF annotation files were used to tag aligned reads. Cellular barcodes and UMIs were directly extracted from Read1. A list of two rounds of cell barcode oligo sequences was used to correct the extracted cellular barcode from read1. Finally, the HTseq package was used to generate digital expression matrices of axolotl[80]. For cell quality control of axolotl data, we excluded cells in which less than 200 transcripts were expressed. In the mixed-species experiment of 3T3 cells and 293T cells, the percentage of uniquely mapping reads for genomes of each species with over 85% UMIs assigned to one species was regarded as species-specific cells, while the remaining cells were labeled collisions. Count matrices of bulk RNA-seq of axolotl specimens were generated by featureCounts[81] with default parameters.

**Cell clustering, visualization, and annotation of CH-RNA-seq data**. Seurat[82] was used for clustering of RNA dataset. The data were $\log_2$ (counts per million (CPM)/100 + 1)-transformed, and a total of 2000 highly variable genes were selected through the "FindVariableFeatures" function by using the "vst" method as inputs for initial principal component analysis (PCA). The number of principal components (PCs) was used for nonlinear dimensionality reduction (t-SNE) and UMAP. We set different resolution parameters between 0.5 and 2.5 in the "FindAllCluster" function and narrowed down to certain cluster numbers by distinguishing differential genes among clusters. These parameters, including the resolution and number of PCs, were adjusted on a per-dataset basis. For batch effect evaluation, axolotl datasets were processed by the "SCTransform" function, and processed data were fed to Scanpy[83]. We chose 50 PCs for PCA and computed the neighborhood graph of cells. We used the R package DoubletFinder[84] to detect and remove the potential doublets. We then used Leiden clustering to cluster cells with a resolution of 2.5 and $k = 15$. We identified 459 subclusters in neotenic axolotls and 304 subclusters in metamorphosed axolotls. Marker genes were calculated by the Wilcoxon rank-sum test. t-SNE was applied to visualize the single-cell transcriptional profile in 2D space of whole datasets for neotenic and metamorphosed axolotls. UMAP was applied to visualize the single-cell transcriptional profile of each organ from neotenic axolotls and metamorphosed axolotls. We adopted SPRING to visualize whole axolotl single-cell RNA datasets from Day 30 to Day 70 post-fertilization[68]. The Wilcoxon rank-sum test was used by running

the "rank_genes_groups" function in Scanpy to find differentially expressed genes in each cluster. We annotated each cell type by marker gene expression patterns with extensive literature reading.

**Single-cell entropy analysis**. We adopted single-cell lineage inference using cell expression similarity and entropy (SLICE) for quantitative measurements of cell differentiation states based on the calculation of single-cell entropy (scEntropy)[37]. We then performed deterministic calculation of scEntropy of individual cells in neotenic and metamorphosed axolotl tissues with default parameters according to the SLICE pipeline.

**Evaluation of cell-type perturbation between neotenic and metamorphosed axolotl skin**. Protocols for perturbation response analysis were followed in the reported study[61]. Briefly, we sampled 11,512 skin cells from neotenic axolotls and merged them with 10,158 metamorphosed axolotl skin cells. Perturbed differentially expressed genes between each cluster were identified using the "FindMarkers" function in Seurat. We used Louvain clustering for both the cell matrix and gene expression modules. Differentially expressed gene modules were used to generate putative perturbed cell types and perturbation gene ontology enrichments.

**Gene regulatory network construction**. The gene regulatory networks of neotenic and metamorphosed axolotls were calculated using bigSCale[66]. For each dataset, we aggregated data from 50 cells in the same cell cluster to make pseudocells for the genetic network. The expression of the pseudocell matrices was fed to bigSCale, and networks were constructed under default parameters. We visualized networks with igraph. The layout of each network is derived from 10,000 iterations of the Fruchterman–Reingold algorithm with nogrid. Cell type annotation is based on the PageRank centrality of node genes.

**MetaNeighbor analysis**. We extracted limb-development-related cell lineages from a whole-organism larval neotenic axolotl dataset. Limb regeneration lineages of the blastema dataset were downloaded in a previous study[18]. To systematically assess the transcriptional similarity between cell types within and between species, we used MetaNeighbor[36] for the merged dataset. Replicability scores (AUROCs) were used to quantify the similarity of cell-type pairs. Hierarchical clustering trees and heatmaps based on AUROC scores were used to reveal the relationships within clusters from two datasets. Hierarchical clustering trees of neotenic and metamorphosed axolotls were constructed with MetaNeighbor with similar strategies.

**Identification of time-associated genes during larval neotenic axolotl limb development**. We identified time-associated genes that showed upregulation and downregulation patterns at both the expression and percentage levels during the 5 larval neotenic axolotl limb developmental points by using Spearman rank correlation analysis in cell types. We calculated the correlation between the gene expression level across 5 time points and the linear vector for each gene. Permutation distribution was used to test the hypothesis of no correlation against the alternative hypothesis of a nonzero correlation to calculate the p value of each gene. In addition to the gene expression, we calculated the percentage of gene expressed in each stage and computed its correlation with the stage vector again for each gene. Genes with a p value < 0.05 were selected at both the expression and percentage levels.

**Axolotl cell landscape website construction**. The main Axolotl Cell Landscape website uses a bootstrap framework to improve overall adaptability and interactivity. Its back end was developed with PHP, R, and MySQL. The main functions of the website are divided into three parts: gallery, landscape, and search. Gallery provides interactive UMAP for datasets from neotenic axolotls, metamorphosed axolotls, and larval axolotls to show the distribution of different clusters. Specific markers for each cluster are listed in Supplementary Data 3. Landscape achieves better visualizations for the global view of major clusters from different datasets. The search describes the expression of a given gene in different clusters from any selected tissue (gene name was demonstrated in original format in axolotl genome annotation file).

**Statistics and reproducibility**. No statistical method was used to predetermine sample size. Randomization and blinding were used. In Fig. 4b, Fig. 7c, Fig. 8g, h,

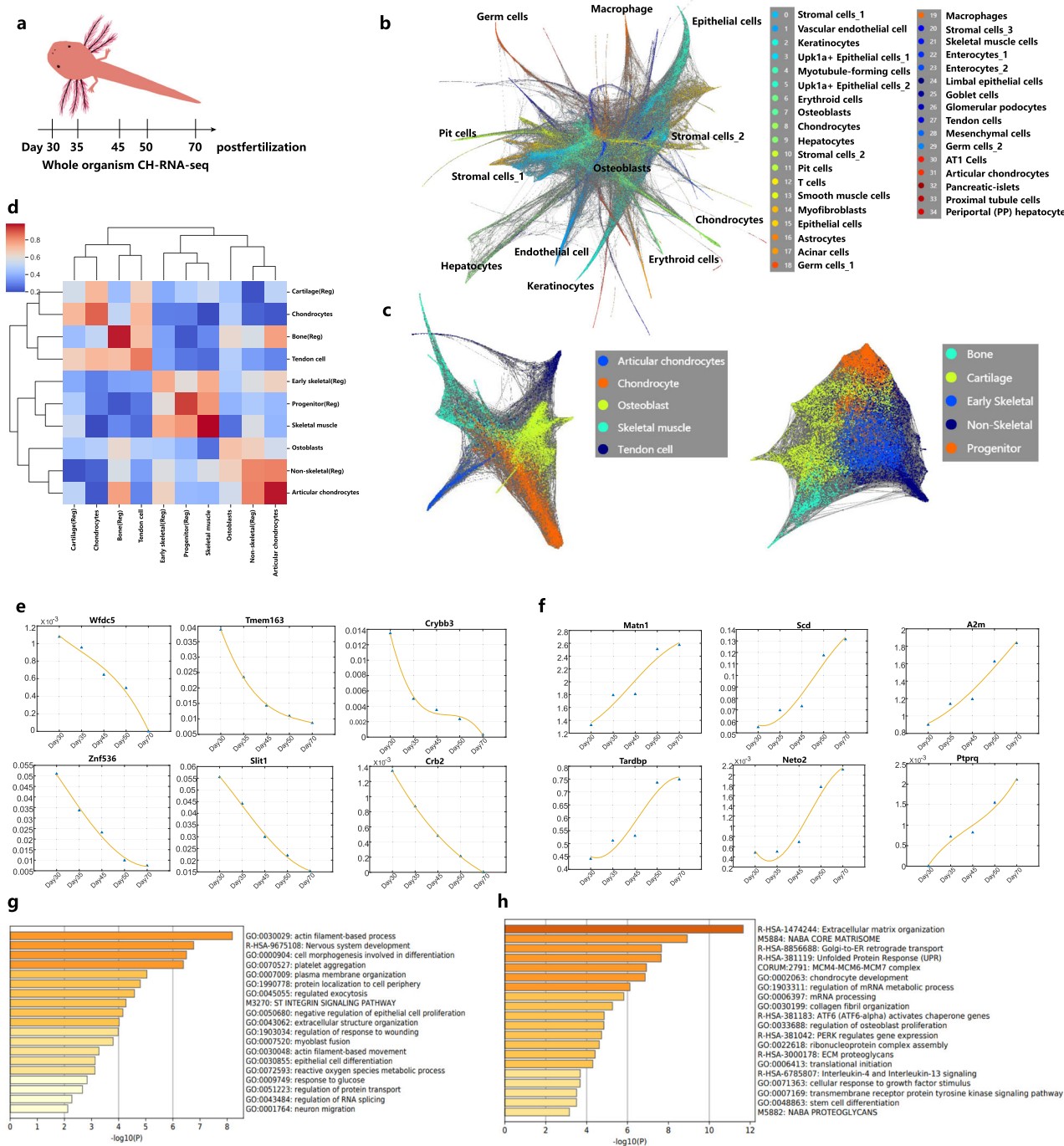

**Fig. 8 Whole-organism CH-RNA-seq mapped neotenic axolotl limb development. a** Workflow for constructing larval neotenic axolotl single-cell landscape using CH-RNA-seq. **b** SPRING visualization of 217,781 single cells of larval neotenic axolotl whole-organism and cell-type annotation. **c** SPRING visualization of limb-development-related cell lineages (left) and limb regeneration blastema cell lineages (right). **d** MetaNeighbor analysis showing the cell-type correlations of limb-development-related cell lineages and limb regeneration blastema cell lineages (Reg). **e** Selected downregulated genes from Day 30 to Day 70 post-fertilization. **f** Selected upregulated genes from Day 30 to Day 70 post-fertilization. Gene ontology (GO) enrichment of downregulated genes (**g**) and upregulated genes (**h**) during limb development (*p* values was calculated by the hypergeometric distribution, statistical test is one-sided, adjustments *p* values were made after *p* value is corrected by Benjamin & Hochberg multiple test).

and Supplementary Fig. 5g, *p* values were calculated by the hypergeometric distribution, statistical test is one-sided, adjustments *p* values were made after *p* value is corrected by Benjamin & Hochberg multiple test. In Fig. 5f–i and Supplementary Fig. 6c–f, *t* test was introduced, adjustments *p* values were made after *p* value is corrected by Benjamin & Hochberg multiple test. In Supplementary Fig. 1g, Mann-Whitney-Wilcoxon test was introduced, adjustments *p* values were made after *p* value is corrected by Benjamin & Hochberg multiple test.

**Reporting summary**. Further information on research design is available in the Nature Research Reporting Summary linked to this article.

## Data availability
Raw data generated by this work have been deposited in the NCBI Gene Expression Omnibus database and are accessible through the following accession numbers:

Bulk RNA-seq, available at "GSE182746 [https://www.ncbi.nlm.nih.gov/geo/query/acc.cgi?acc=GSE182746]", scRNA-seq, available at "GSE201446 [https://www.ncbi.nlm.nih.gov/geo/query/acc.cgi?acc=GSE201446]". Processed scRNA-seq data have been deposited on Figshare and are accessible through the following sites: "scRNA-seq data from adult axolotls [https://figshare.com/s/8e22f99511ef2ddfd7f3]" and "scRNA-seq data from larval-stage axolotls [https://figshare.com/s/aa6ad78b28a205a87e56]". scRNA-seq data can also be accessed on the "Axolotl Cell Atlas [http://bis.zju.edu.cn/ACA/]". Previously published single-cell RNA-seq data that are associated with Fig. 1 are available in the GEO under the accession codes: sci-RNA-seq, available at "GSE98561", SPLiT-seq, available at "GSE110823", Drop-seq, available at "GSE63269" and 10X Genomics sc-RNA-seq,1k_hgmm_v3_nextgem dataset, available at "10X Genomics [https://support.10xgenomics.com/single-cell-gene-expression/datasets/3.0.2/1k_hgmm_v3_nextgem?]". Public axolotl genome and transcriptome datasets used in this study: "Axolotl genome [https://www.axolotl-omics.org/assemblies]", Axolotl transcriptome, available at "GSE92429" and NCBI SRA repository, available at "SRP093628". Public transcriptome datasets of *Ambystoma velasci* used in this study: NCBI BioProject available at "RJNA557269". All other relevant data supporting the key findings of this study are available within the article and its Supplementary Information files or from the corresponding author upon reasonable request.

## Code availability

Custom scripts for data analysis in this study were present in "Zenodo [https://doi.org/10.5281/zenodo.6698785]".

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

## Acknowledgements

We thank Mingyong Zhou, Xing Fang, Yuan Liao, Junqing Wu, Renying Wang, Yuqing Mei, Xueyi Wang, Haofu Niu, Tingyue Zhang, Mengmeng Jiang, Huiyu Sun, Danmei Jia, and Jiangxue Huang for support on the project. We thank the Center of Cryo-Electron Microscopy (CCEM) at Zhejiang University for the resources on the computation. Thanks for the technical support by the core facilities of Zhejiang University Medical Center and Liangzhu Laboratory. Thanks for the technical support by the Core Facilities, Zhejiang University School of Medicine. This work was supported by the National Natural Science Foundation of China (grants 32001068, 31930028, 31871473, 31922049, 91842301, G.G.), the National Key Research and Development Program (grants 2018YFA0800503, 2018YFA0107804, 2018YFA0107801, G.G.), and the Fundamental Research Funds for the Central Universities (G.G.).

## Author contributions

The project was conceived by G.G. Tissue digestion experiments were performed by G.Z., F.Y., H.C. Experiments were performed by G.Z., F.Y., H.C, L.Y., X.W., Bioinformatics analysis was performed by W.E., C.Y., Z.S., L.F., Q.G., Y.X., L.M., J.L., Y.F.. P.Z., J.W., W.E., Y.Z., M.C. performed website construction. J.C.P., A.C.R, S.F., J.F., D.G., S.X. provided supports on animal housing and data analysis. The paper was written by G.G., X.H., F.Y. and funding was acquired by G.G. and X.H.

## Competing interests

The authors declare no competing interests.

## Additional information

 

