## [Peer Review File · Nature Communications]

REVIEWER COMMENTS

Reviewer #1 (Remarks to the Author):

In this manuscript Ye and colleagues describe a modified SPLIT_Seq-like protocol called CH-Seq for massive parallel single cell sequencing and apply it to profiling cells organism wide comparing pre-metamorphic and post-metamorphic axolotls. The CH-eq method, oligo hybridization rather than ligation of adapters is used to incorporate the second round of barcodes, yielding a high diversity of barcodes. The authors benchmark the method on mouse and human cells, comparing the depth of their data to published data of other major single cell sequencing methods. The authors then go on to sequence millions of cells from the two types of axolotls, cluster cell types, derive differentially expressed genes across tissues, and build gene regulatory networks. They point out many potentially interesting differences between pre and post metamorphic axolotl tissues.

This work presents potentially very valuable dataset for the understanding of cell type composition and molecular features of pre and post metamorphic tissues. Overall, their method and their dataset requires substantial further validation, and their biological observations also require more indepth treatment to arrive at a level appropriate for Nature Communications.

1. Characterization of human and mouse datasets. The authors nicely provide a table and box plots comparing the number of UMIs and unique genes identified by the different methods and their method appears to be comparable. In figure 1b, they refer to comparison to Encode datasets, and show a mapping profile for one location to the human genome. It would be very valuable for the authors to find genome-wide means to compare the mapped profiles of reads to the genome between the different methods.

2. The axolotl dataset was sequenced very shallowly, and for some reason it seems that a very low percentage of the reads showed unique mappability. Therefore the number of reads per cell is rather low. Can the authors provide an explanation for this? In this case, considering that they sequenced many cells, and perform cluster analysis, it would be important to define the number of cells per cluster, and how many unique genes were identified per cluster.

3. The authors compare their single cell tissue-specific data to bulk RNASequencing across tissues. I am worried that in a number of cases, the cognate tissue in the bulk dataset is not the best match to the single cell dataset. Is that because the number of gene identified in the two datasets is different, or

because for example, tissue dissociation in the single cell protocol yields an uneven distribution of cell types?

4. In terms of comparing pre and post metamorphic tissues, I am concerned that if there is a general change in metabolism, and for example, the total levels of RNA per cell changes upon metamorphosis, that some of the apparently differential expression could be due to differences in transcript amplification due to such a general change in cell physiology. Also, the authors state that certain cell types are present or absent specifically in one stage--but if there is a change in tissue dissociation characteristics, it could be that such cells are lost preferentially in one sample versus another. Many of these issues, could be addressed by following up on the data with quantitative in situ hybridization at the tissue level.

5. In general, the conclusions being drawn about the presence or absence of expression of certain genes needs to be followed up by in situ hybridization methods.

6. In figure 3E, the authors visualize differentially expressed genes among tissues in pre and post metamorphic animals. I think it would be more interesting to visualize genes that are differentially expressed between pre and post-metamorphic stages in a tissue specific manner.

7. The manuscript contains a wealth of potentially interesting observations. Much of the description of potentially interesting findings needs to be described in in-depth quantitative terms. It would also helpful if the manuscript focuses one or a few major story-lines.

Reviewer #2 (Remarks to the Author):

The manuscript entitled: "The single-cell transcriptional landscape of the axolotl" by Fang Ye et al., authors describe the transcriptional dynamics of genes from 19 tissues of either neotenic or fully metamorphosed *Ambystoma mexicanum* organisms.

In recent years several RNA-Seq approaches to profile diverse tissues, organs and processes have been reported for neotenic and metamorphosed axolotl or close species, either via traditional RNA-seq experiments or novel single-cell techniques (Bryant et al., 2018, Gerber et al., 2018., Caballero-Perez, et al., 2018; Palacios-Martinez, et al., 2020).

The addition of the work by Fang ye and collaborators to the field is important, since they use Single-cell techniques to cover up to 19 tissues in neotenic and metamorphosed *A. mexicanum* organisms.

Although one can claim lack of originality in the final goal of this study, it provides data for novel tissues at a more detailed level than the previously mentioned approaches. However one concern is that this study does neither acknowledges properly previous RNA-Seq studies, nor uses transcriptional published data to make detailed comparisons with their own. One would expect that authors could go deeper in the analyses, specially since authors attempt to publish in Nature Communications.

Also, there are several claims along the text that are not sufficiently supported. Prior acceptance and publication several comments and suggestions need to be revised. All the detailed suggestions, comments and critics are highlighted in the attached PDF file and specific comment boxes related to each of these observations can be consulted by the Editor and authors.

The comparison between the obtained transcriptional profiles with previous RNA-seq studies is quite poor. The discussion is poor too. Authors should enlist at the genetic level which are the novel findings of this study that have been absent from previous RNA-seq studies and discuss them properly.

One analysis that could enrich and improve the manuscripts is a comparative analyses among the DEGs observed in metamorphosed *A. mexicanum* tissues versus DEGs of metamorphosed *A. velasci* tissues, this may be done in order to confirm a subset of transcripts associated with fully metamorphic state among two quite related *Ambystoma* species.

Please revise each comment box and provide responses and changes to them.

Reviewer #3 (Remarks to the Author):

The authors proposed a single-cell RNA-seq technique by combinatorial barcoding and generated a single-cell atlas of axolotl in development. The dataset is featured with over 1 million single cells across primary tissues in neotenic and metamorphosed axolotls. They characterized cell-type-specific gene signatures and analyzed dynamic gene expression patterns during limb development. The dataset could be helpful for exploring the molecular identity of cells involved in axolotl development. There are several major concerns, especially about the quality of the dataset, as discussed below.

1. The technique uses a very similar cell fixation and barcoding workflow as the published single-cell RNA-seq techniques by combinatorial indexing. Also, it is not obvious to see much improvement in throughput, efficiency, or any new information that can be recovered from the strategy. It is more like an optimized version of the current techniques instead of a new strategy as proposed in the abstract.

2. Fig. 1C. For comparing different techniques, the authors should sample the same number of reads per cell. Also, it is not convincing to compare the signal from single-cell RNA-seq with single-nucleus RNA-seq.

3. Based on Fig. S1C, there is a strong batch effect between different individuals in both neotenic and metamorphosed axolotls. This batch effect should be removed before downstream analysis.

4. It is critical to ensure that the batch effect does not interfere with the downstream sub-clustering analysis.

4. Page 7, line 141: "Approximately 20% of cells in the library ultimately passed filtration steps". This is a concern about the quality of the dataset. Why are 80% of cells lost during the filtration step?

5. It seems some clusters overlap with each other based on the UMAP plot in the sub-cluster analysis (e.g., WT_Gill) but are assigned to different names. This should be clarified.

6. Line 289. "Umod was downregulated in metamorphosed axolotl skin." This conclusion is not apparent based on the plot.

7. The authors claimed that Chga+ cells were detected only in the neotenic heart. However, this could be simply due to the higher number of cells profiled in the neotenic heart.

8. Several claims in the manuscript lack support from figures (e.g., the conclusion in line 187, line 202). These should be fixed together with some obvious grammar errors across the manuscript.

REVIEWER COMMENTS

Reviewer #1 (Remarks to the Author):

In this manuscript Ye and colleagues describe a modified SPLIT_Seq-like protocol called CH-Seq for massive parallel single cell sequencing and apply it to profiling cells organism wide comparing pre-metamorphic and post-metamorphic axolotls. The CH-seq method, oligo hybridization rather than ligation of adapters is used to incorporate the second round of barcodes, yielding a high diversity of barcodes. The authors benchmark the method on mouse and human cells, comparing the depth of their data to published data of other major single cell sequencing methods. The authors then go on to sequence millions of cells from the two types of axolotls, cluster cell types, derive differentially expressed genes across tissues, and build gene regulatory networks. They point out many potentially interesting differences between pre and post metamorphic axolotl tissues.

This work presents potentially very valuable dataset for the understanding of cell type composition and molecular features of pre and post metamorphic tissues. Overall, their method and their dataset require substantial further validation, and their biological observations also require more in-depth treatment to arrive at a level appropriate for Nature Communications.

1. Characterization of human and mouse datasets. The authors nicely provide a table and box plots comparing the number of UMIs and unique genes identified by the different methods and their method appears to be comparable. In figure 1b, they refer to comparison to Encode datasets, and show a mapping profile for one location to the human genome. It would be very valuable for the authors to find genome-wide means to compare the mapped profiles of reads to the genome between the different methods.

2. The axolotl dataset was sequenced very shallowly, and for some reason it seems that a very low percentage of the reads showed unique map ability. Therefore, the number of reads per cell is rather low. Can the authors provide an explanation for this? In this case, considering that they sequenced many cells, and perform cluster analysis, it would be important to define the number of cells per cluster, and how many unique genes were identified per cluster.

3. The authors compare their single cell tissue-specific data to bulk RNA sequencing across tissues. I am worried that in a number of cases, the cognate tissue in the bulk dataset is not the best match to the single cell dataset. Is that because the number of genes identified in the two datasets is different, or because for example, tissue dissociation in the single cell protocol yields an uneven distribution of cell types?

4. In terms of comparing pre and post metamorphic tissues, I am concerned that

if there is a general change in metabolism, and for example, the total levels of RNA per cell changes upon metamorphosis, that some of the apparently differential expression could be due to differences in transcript amplification due to such a general change in cell physiology. Also, the authors state that certain cell types are present or absent specifically in one stage—but if there is a change in tissue dissociation characteristics, it could be that such cells are lost preferentially in one sample versus another. Many of these issues, could be addressed by following up on the data with quantitative in situ hybridization at the tissue level.

5. In general, the conclusions being drawn about the presence or absence of expression of certain genes needs to be followed up by in situ hybridization methods.

6. In figure 3E, the authors visualize differentially expressed genes among tissues in pre and post metamorphic animals. I think it would be more interesting to visualize genes that are differentially expressed between pre and post-metamorphic stages in a tissue specific manner.

7. The manuscript contains a wealth of potentially interesting observations. Much of the description of potentially interesting findings needs to be described in in-depth quantitative terms. It would also helpful if the manuscript focuses one or a few major story-lines.

Reviewer #2 (Remarks to the Author):

The manuscript entitled: "The single-cell transcriptional landscape of the axolotl" by Fang Ye et al., authors describe the transcriptional dynamics of genes from 19 tissues of either neotenic or fully metamorphosed *Ambystoma mexicanum* organisms.

In recent years several RNA-Seq approaches to profile diverse tissues, organs and processes have been reported for neotenic and metamorphosed axolotl or close species, either via traditional RNA-seq experiments or novel single-cell techniques (Bryant et al., 2018, Gerber et al., 2018., Caballero-Perez, et al., 2018; Palacios-Martinez, et al., 2020).

The addition of the work by Fang ye and collaborators to the field is important, since they use Single-cell techniques to cover up to 19 tissues in neotenic and metamorphosed *A. mexicanum* organisms.

Although one can claim lack of originality in the final goal of this study, it

provides data for novel tissues at a more detailed level than the previously mentioned approaches. However one concern is that this study does neither acknowledge properly previous RNA-Seq studies, nor uses transcriptional published data to make detailed comparisons with their own. One would expect that authors could go deeper in the analyses, specially since authors attempt to publish in Nature Communications.

Also, there are several claims along the text that are not sufficiently supported. Prior acceptance and publication several comments and suggestions need to be revised. All the detailed suggestions, comments and critics are highlighted in the attached PDF file and specific comment boxes related to each of these observations can be consulted by the Editor and authors.

The comparison between the obtained transcriptional profiles with previous RNA-seq studies is quite poor. The discussion is poor too. Authors should enlist at the genetic level which are the novel findings of this study that have been absent from previous RNA-seq studies and discuss them properly.

One analysis that could enrich and improve the manuscripts is a comparative analyses among the DEGs observed in metamorphosed *A. mexicanum* tissues versus DEGs of metamorphosed *A. velasci* tissues, this may be done in order to confirm a subset of transcripts associated with fully metamorphic state among two quite related *Ambystoma* species.

Please revise each comment box and provide responses and changes to them.

Reviewer #3 (Remarks to the Author):

The authors proposed a single-cell RNA-seq technique by combinatorial barcoding and generated a single-cell atlas of axolotl in development. The dataset is featured with over 1 million single cells across primary tissues in neotenic and metamorphosed axolotls. They characterized cell-type-specific gene signatures and analyzed dynamic gene expression patterns during limb development. The dataset could be helpful for exploring the molecular identity of cells involved in axolotl development. There are several major concerns, especially about the quality of the dataset, as discussed below.

1. The technique uses a very similar cell fixation and barcoding workflow as the published single-cell RNA-seq techniques by combinatorial indexing. Also, it is not obvious to see much improvement in throughput, efficiency, or any new information that can be recovered from the strategy. It is more like an optimized version of the current techniques instead of a new strategy as proposed in the

abstract.

2. Fig. 1C. For comparing different techniques, the authors should sample the same number of reads per cell. Also, it is not convincing to compare the signal from single-cell RNA-seq with single-nucleus RNA-seq.

3. Based on Fig. S1C, there is a strong batch effect between different individuals in both neotenic and metamorphosed axolotls. This batch effect should be removed before downstream analysis.

4. It is critical to ensure that the batch effect does not interfere with the downstream sub-clustering analysis.

4. Page 7, line 141: "Approximately 20% of cells in the library ultimately passed filtration steps". This is a concern about the quality of the dataset. Why are 80% of cells lost during the filtration step?

5. It seems some clusters overlap with each other based on the UMAP plot in the sub-cluster analysis (e.g., WT_Gill) but are assigned to different names. This should be clarified.

6. Line 289. "Umod was downregulated in metamorphosed axolotl skin." This conclusion is not apparent based on the plot.

7. The authors claimed that Chga⁺ cells were detected only in the neotenic heart. However, this could be simply due to the higher number of cells profiled in the neotenic heart.

8. Several claims in the manuscript lack support from figures (e.g., the conclusion in line 187, line 202). These should be fixed together with some obvious grammar errors across the manuscript.

Responses to the reviewers' comments (NCOMMS-21-39949)

Below is our point-by-point response to reviewers' comments. The criticisms are in **RED**, and our responses are in **BLUE**.

We sincerely thank all the reviewers for the valuable comments and suggestions that help us to improve the study. We have revised our manuscript accordingly and provided a detailed point-by point response to the comments below.

Point-by-point response

Reviewer #1 (Remarks to the Author):

In this manuscript Ye and colleagues describe a modified SPLIT_Seq-like protocol called CH-Seq for massive parallel single cell sequencing and apply it to profiling cells organism wide comparing pre-metamorphic and post-metamorphic axolotls. The CH-eq method, oligo hybridization rather than ligation of adapters is used to incorporate the second round of barcodes, yielding a high diversity of barcodes. The authors benchmark the method on mouse and human cells, comparing the depth of their data to published data of other major single cell sequencing methods. The authors then go on to sequence millions of cells from the two types of axolotls, cluster cell types, derive differentially expressed genes across tissues, and build gene regulatory networks. They point out many potentially interesting differences between pre and post metamorphic axolotl tissues.

This work presents potentially very valuable dataset for the understanding of cell type composition and molecular features of pre and post metamorphic tissues.

Overall, their method and their dataset require substantial further validation, and their biological observations also require more indepth treatment to arrive at a level appropriate for Nature Communications.

1. Characterization of human and mouse datasets. The authors nicely provide a table and box plots comparing the number of UMIs and unique genes identified by the different methods and their method appears to be comparable. In figure 1b, they refer to comparison to Encode datasets, and show a mapping profile for one location to the human genome. It would be very valuable for the authors to find genome-wide means to compare the mapped profiles of reads to the genome between the different methods.

Response: We thank the reviewer for raising this point. The representative genome browser view of genome-wide reads tracks in single-cell methods and ENCODE data portal could display the enrichment of reads around transcription termination sites (TTSs) in order to evaluate the effectiveness of the methods. In revised Figure 1B, we have compared CH-seq with representative scRNA-seq method (10X Genomics scRNA-seq, Version 3). In revised text, we have included following conclusions in line 135-139: “Genome read coverage from CH-seq showed a high correlation with published ENCODE data. We compared CH-seq with other representative scRNA-seq methods. RNA reads in CH-RNA-seq libraries were enriched at upstream regions of transcription termination sites (TTSs) (Figure 1B)”. As shown in Figure 1B, data quality in CH-seq is comparable with representative commercialized method and public dataset.

2. The axolotl dataset was sequenced very shallowly, and for some reason it seems that a very low percentage of the reads showed unique mappability. Therefore, the number of reads per cell is rather low. Can the authors provide an explanation for this? In this case, considering that they sequenced many cells, and perform cluster analysis, it would be important to define the number of cells per cluster, and how many unique genes were identified per cluster.

Response: We thank the reviewer for bringing up this point. Due to the limitation of sequencing costs and number of cells in the sequencing libraries, the axolotl dataset was sequenced very shallowly. In pool-split procedures, number of cells input in each sub-library (labeled using a i7 index) is about 5000. We sequenced 96 sub-libraries in a single chip on MGISEQ-T7 which generated about 3,500M total reads. After quality control of raw sequencing reads, the average number of raw reads in each sub-library is about 20M. Overall alignment rate of raw reads in each library is around 90% (45% reads aligned exactly 1 time; 45% reads aligned >1 time; 10% reads aligned 0 times). The mapping rate is reasonable, but the raw reads are not enough for such number of cells. Each sub-library generated around 2,000 cells passed transcripts number cutoff value. Limited sequencing depth and too much number of cells in the sequencing library lead to the low number of

reads per cell. It is flexible to seq more reads and pool less sub-libraries. The advantage of shallow sequencing is to cover more cells with less cost, make it affordable for most labs. On the other hand, we demonstrate the feasibility of pool-split single-cell RNA-seq strategy on adult samples, while other methods (sci-RNA-seq1,3; SPLiT-seq) were focused on embryonic or fetal samples.

In order to define the unique genes identified per cluster, we analyzed differentially expressed genes in each tissue and merged datasets of all tissues (see Figure S2). We observed slightly differences in unique genes number between neotenic axolotl and metamorphosed axolotl datasets. Despite the limitation of sequencing depth, we detected sufficient unique genes for down-stream analysis.

3. The authors compare their single cell tissue-specific data to bulk RNA Sequencing across tissues. I am worried that in a number of cases, the cognate tissue in the bulk dataset is not the best match to the single cell dataset. Is that because the number of genes identified in the two datasets is different, or because for example, tissue dissociation in the single cell protocol yields an uneven distribution of cell types?

Response: We analyzed all the genes detected in our bulk RNA-seq data and single-cell RNA-seq data. Generally, most genes were overlapped between two datasets (See Figure 1 for reviewer 1). And indeed, tissue dissociation in the single cell protocol introduced unbalanced distribution of cell types. Unbiased tissue dissociation protocols generated large number of epithelial cells in each tissue, which led to high gene expression correlations between skin, limbs, tail and gill. We observed specific correlations in the same tissue between bulk data and single-cell data. Epithelial cells across different tissues exhibit high gene expression similarity. We believe further single-nuclei CH-seq will solve the tissue dissociation bias.

Figure 1 for Reviewer 1. Venn diagrams of overlapped genes between bulk RNA-seq data and single-cell RNA-seq data (Left: Datasets of neotenic axolotls; Right: Datasets of metamorphosed axolotls).

4. In terms of comparing pre and post metamorphic tissues, I am concerned that if there is a general change in metabolism, and for example, the total levels of RNA per cell changes upon metamorphosis, that some of the apparently differential expression could be due to differences in transcript amplification due to such a general change in cell physiology. Also, the authors state that certain cell

types are present or absent specifically in one stage--but if there is a change in tissue dissociation characteristics, it could be that such cells are lost preferentially in one sample versus another. Many of these issues, could be addressed by following up on the data with quantitative in situ hybridization at the tissue level.

Response: To address reviewer' s questions, we searched KEGG PATHWAY Database (<https://www.genome.jp/kegg/pathway.html#metabolism>) for gene modules involved in metabolic process and transcription process. Then we compared gene module expression levels between neotenic axolotls and metamorphosed axolotls in our single-cell datasets (using AddModuleScore()) function in Seurat (Satija, Farrell et al. 2015)). The results showed that in metamorphosed axolotl, most general metabolic process, including carbohydrate metabolism, amino acid metabolism, glycan biosynthesis and metabolism demonstrate down-regulation in varying degrees (Figure 2 below). In a protein expression study of neotenic and metamorphosed axolotl limb regeneration, KEGG pathway analyses also showed that multiple metabolic process were mostly enriched in down-regulated proteins in metamorphic samples (Sibai, Altuntas et al. 2020). We observed higher level of translation, RNA degradation and transcription module score in metamorphosed axolotls.

We noticed tissue dissociation bias in single-cell RNA-seq may that may cause the loss of rare cell types. To exclude the possibility that the identified differentially expressed genes is from dissociation bias, we performed RNA *in situ* hybridization of perturbed genes in single-cell RNA-seq datasets (See Figure 3, Figure 6). In this part, we have revised descriptions of these differentially expressed genes based on the dot plots and RNA *in situ* hybridization results (line 227-278).

Figure 2 for Reviewer 1. Representative RNA in situ hybridizations of neotenic axolotl tissues (NEO) and metamorphosed axolotl tissues (META). $n = 2$ animals per probe (Rep). All green scale bars are $25 \mu\text{m}$.

Figure 3 for Reviewer 1. Boxplots of gene module expression levels involved in metabolic and transcription process. Functional gene modules were obtained from KEGG PATHWAY Database (<https://www.genome.jp/kegg/pathway.html#metabolism>).

5. In general, the conclusions being drawn about the presence or absence of expression of certain genes needs to be followed up by *in situ* hybridization methods.

Response: In the revised manuscript, we have added dot plots and RNA *in situ* hybridization results to support the conclusions of perturbed cell types in neotenic axolotls and metamorphosed axolotls (Figure 3, Figure 6). Generally, all the genes demonstrate different expression levels or ratio between two stages (Krt6a, Chga, Mmp19, Muc5ac, Muc5b, Cd109, Muc4). We modified the claims about other genes which are not “absence” but showed differentially expression patterns.

6. In figure 3E, the authors visualize differentially expressed genes among tissues in pre and post metamorphic animals. I think it would be more interesting to visualize genes that are differentially expressed between pre and post-metamorphic stages in a tissue specific manner.

Response: We appreciate the reviewer's suggestion. We have visualized tissue specific differentially expressed genes between pre and post-metamorphic stage (Figure 4A). We also performed function enrichment of those differentially expressed genes in each tissue (Figure 4B). In this new part (line 280-320), we also compared differentially expressed genes in heart and lung in axolotl and *Ambystoma velasci* (*A. velasci*) (Figures S5F and S5G). Conserved signatures in two species (e.g. Gata4 in heart) were identified. Interestingly, we observed enrichment of fibrinogens in metamorphosed axolotl liver and prostaglandins in stomach. These results revealed the enhancement of tissue function after metamorphosis.

7. The manuscript contains a wealth of potentially interesting observations. Much of the description of potentially interesting findings needs to be described in in-depth quantitative terms. It would also helpful if the manuscript focuses one or a few major story-lines.

Response: We thank the reviewer for these comments. In this work, we introduced CH-seq as a reliable single-cell sequencing method. We focused on the comparison of single-cell transcriptome between neotenic axolotl and metamorphosed axolotl. In the revised manuscript, we analyzed tissue-specific differentially expressed genes and related perturbed cell types and discussed their potential function in metamorphosis. We consider perturbed genes in skin and limbs epithelial cells (Krt6a, Muc4) after tissue remodeling were associated with cell type perturbation in skin. We observed regulons related to perturbed cell types may not recognized as the top differentially expressed genes. We further constructed gene regulatory networks and compared driven regulons in neotenic and metamorphic axolotls. Stronger relationship between stromal and epithelial cells was observed in the metamorphosed axolotl. We also identified key regulons associated with enhancement of stromal and epithelial cells function in metamorphosed axolotl. The limb regeneration of axolotl has been well studied. In this case, we mapped cell landscape of larval stage axolotl (limb development) as a supplement resource of axolotl cell landscape in the last part of the study.

In the part of larval stage axolotl, we performed a time-dependent gene expression analysis related in axolotl limb development and identified conserved gene signatures such as Matn1 and A2m. In the present work, we focused primarily on method and cell landscape database construction, with less investigation of cell perturbation mechanisms in metamorphosis due to limited space. We expect there to be great interest in dissecting the specific characteristics of respiratory system (lung, skin, gill) and circulatory system (Heart) in axolotl metamorphosis, which will be helpful for exploring novel mechanisms in evolution and regeneration. Our future ATAC-seq study of multi-species will discuss these problems based on cross species comparison.

Reviewer #2 (Remarks to the Author):

The manuscript entitled: "The single-cell transcriptional landscape of the axolotl" by Fang Ye et al., authors describe the transcriptional dynamics of genes from 19 tissues of either neotenic or fully metamorphosed *Ambystoma mexicanum* organisms.

In recent years several RNA-Seq approaches to profile diverse tissues, organs and processes have been reported for neotenic and metamorphosed axolotl or close species, either via traditional RNA-seq experiments or novel single-cell techniques (Bryant et al., 2018, Gerber et al., 2018., Caballero-Perez, et al., 2018; Palacios-Martinez, et al., 2020).

The addition of the work by Fang ye and collaborators to the field is important, since they use Single-cell techniques to cover up to 19 tissues in neotenic and metamorphosed *A. mexicanum* organisms.

Although one can claim lack of originality in the final goal of this study, it provides data for novel tissues at a more detailed level than the previously mentioned approaches. However, one concern is that this study does neither acknowledges properly previous RNA-Seq studies, nor uses transcriptional published data to make detailed comparisons with their own. One would expect that authors could go deeper in the analyses, especially since authors attempt to publish in Nature Communications.

Also, there are several claims along the text that are not sufficiently supported. Prior acceptance and publication several comments and suggestions need to be revised. All the detailed suggestions, comments and critics are highlighted in the attached PDF file and specific comment boxes related to each of these observations can be consulted by the Editor and authors.

The comparison between the obtained transcriptional profiles with previous RNA-seq studies is quite poor. The discussion is poor too. Authors should enlist at the genetic level which are the novel findings of this study that have been absent from previous RNA-seq studies and discuss them properly.

One analysis that could enrich and improve the manuscripts is a comparative analysis among the DEGs observed in metamorphosed *A. mexicanum* tissues versus DEGs of metamorphosed *A. velasci* tissues, this may be done in order to confirm a subset of transcripts associated with fully metamorphic state among two quite related *Ambystoma* species.

Please revise each comment box and provide responses and changes to them.

Response: We sincerely thank the reviewer for these remarks, which we believe will improve the quality and accessibility of our study. We also thank the reviewer for alerting us to these publications. In our original manuscript, we mentioned previous transcriptome, genome, proteome and histology studies in the introduction. Thus, in our revised version, we acknowledged previous RNA-Seq studies, genome studies. We made comparison between published RNA-seq studies and our single-cell data and discussed them in the results. We also compared differentially expressed genes in heart and lung between metamorphic stage in axolotl and *Ambystoma velasci* (*A. velasci*). We found high similarity between bulk tissue enriched transcripts and differentially expressed genes in single-cell datasets (line 201-216; line 305-315). We will provide the detailed response to the comment boxes below.

1. Line 36: *“Is not promising, it is a well-established model”*

Response: We have now revised and corrected description in the updated manuscript.

2. Line 46: *“...revealed the heterogeneity of structural cells in different tissues and established their regulatory network.”*

Response: We have now revised and corrected the word “structural cells”. We use the word “structural cells” from a published work (Krausgruber, Fortelny et al. 2020) without a certain definition. We apologize for our mistake. We have changed the word “structural cells” to “non-immune parenchymal cells” in the revised manuscript. Single-cell regulatory network of major parenchymal cell types refer to the global gene regulatory networks analysis in all the tissues (Line 409-438, Figure 7), we found enriched function of collagen fibril organization, muscle structure development and tissue morphogenesis in metamorphosed axolotl gene regulatory network.

3. Line 54-57: *“I do not think that regeneration capacity has been tested for almost all organs in axolotl, if so, please cite the articles, otherwise refer only to the tested regenerative organs.”*

Response: We thank the reviewer for pointing this out. We have revised the descriptions and listed the reported organs in published axolotl regeneration studies. We cited a representative review due to the limited number of references (line 59-61). Other related published works were listed below (Lung, gill (Jensen, Giunta et al. 2018, Cadiz and Jonz 2020)).

4. *Line 58: “The function of a key is not to break down barriers but to open doors. Analogy is wrong.”*

Response: We thank the reviewer for noticing this mistake. We have now corrected it in the revised version.

5. *Line 79: “de novo (cursives)”*

Response: We have corrected the mistake here.

6. *Line 99-104: “Authors should acknowledge previous efforts in tissues and organs of adult axolotls with different strategies and make clear the differences with their own strategy (Caballero-Perez et al., 2018; Byant et al, 2018).”*

Response: We have now revised this part as requested. We acknowledged previous transcriptome studies using TRIZOL reagent to extract total RNA in tissues and described our single-cell strategy. We further discussed and compared their results below (Line 201-216).

7. *Line 118: “in situ (cursives)”*

Response: We have corrected the mistake here.

8. *Line 183-194: “This is a strong statement; how can neotenic tissues be in a stable state and lose transcriptional plasticity in comparison with the metamorphosed organism? If the samples of metamorphosed organisms were collected once the process occurred, why the talk about transcriptional plasticity during metamorphosis, if the claim in method they collected tissues from fully metamorphosed organism? This should be either strengthen with mire evidence or change the claiming.”*

Response: We thank the reviewer for pointing out this observation. Indeed, we used an inappropriate strong statement in the original version. Single-cell entropy evaluation can only give a relativity result between two stages. In *Ambystoma velasci*, fully metamorphosed stage exhibited more differentially expressed genes than metamorphic climax stage. Transcriptional plasticity in entropy algorithm is correlated with number of differentially expressed genes. In this case, we changed the claim about “stable state” in neotenic axolotl. Single-cell entropy results suggested higher transcriptional plasticity in metamorphosis stage (Line 176-186).

9. *Line 196: “How these transcriptional subsets correlate with those in Bryant*

etal and Calallero-Pérez etal? How these transcriptome subtype correlates with those of Palacios-Martinez etal, 2018. ”

Response: We analyzed tissue specific genes in wild-type neotenic axolotls (*Calallero-Pérez etal, 2018*) and d/d *A. mexicanum* strain (*Bryant etal, 2017*). We observed high correlation between tissue enriched transcriptome and our single-cell transcriptome datasets (Figures S1A). We discussed correlation of these tissue enriched transcripts with our single-cell clusters (Line 201-216). Tissue-specific transcripts in these works could match the results in single-cell subsets. We also observed some different expression patterns in single-cell datasets. For example, in bulk RNA-seq study (*Bryant etal, 2017*), bone-enriched marker, cathepsin k (Ctsk), is highly expressed in ossified portions of the humerus, but we found Ctsk demonstrates higher expression level in macrophages clusters located in bone microenvironment (Herroon, Rajagurubandara et al. 2013). We also discussed differentially expressed genes in metamorphosed *A. velasci* lung, gill and heart (*Palacios-Martinez etal, 2020*) and compared their function enrichment terms with our single-cell datasets (Line 305-315; Figures S5F and S5G).

10. Line 225-227: *“A more closed species to Axolotl than Xenopus has been focus of study during metamorphosis. A close relative to A. mexicanum (A. velasci) from the same genus has been used to study organ remodeling transcriptomics during metamorphosis, this study should be acknowledged and used instead the Xenopus example. ”*

Response: We agree with this reviewer’s comment. We have acknowledged and cited *A. velasci* study (*Palacios-Martinez etal, 2020*) instead of the *Xenopus* example.

11. Line 411: *“should be bold letters”*

Response: We have now corrected it in the revised version.

12. Line 489-497: *“Again, there is a gap between the genome and this study, which is the recognition and comparisson of several RNAseq approaches previously published in axolotl. They should be dicussed and compared here. ”*

Response: We thank the reviewer for raising this point. In the revised version, we discussed representative axolotl RNA-seq studies and their novel findings (Line 488-494). Then, we introduced our single-cell landscape of larva axolotl limb development as another valuable reference in the field.

13. Line 546: *“Please enlist at the genetic level which are the novel findings of this study that have been absent from previous RNA Seq and discuss them. ”*

Response: We thank reviewer for this comment. In the revised version, we summarized findings in cell type perturbations and gene functions between neotenic axolotl and metamorphosed axolotl at single-cell level (line 544–561). Generally, single cell landscape of neotenic, metamorphosed and larval stage axolotl could serve as resources in future study of axolotl as well as cross-species comparison. Tissue-based single-cell resolution datasets could associate differentially expressed genes with certain cell types (line 227–278, Figure3). Differential genes expression analysis of axolotl tissues after metamorphosis revealed other perturbed genes and their function in tissue remodeling (line 280–320). From another point of view, single-cell gene regulatory network of driven regulons also provides novel insights beyond the “up” and “down” regulation of target genes expression (Figure 7). In this case, our further epigenetic studies and multi-omics validation (metabonomic, proteomic, ATAC-seq) of axolotl and other related species will reveal the mechanisms in neoteny and regeneration.

Reviewer #3 (Remarks to the Author):

The authors proposed a single-cell RNA-seq technique by combinatorial barcoding and generated a single-cell atlas of axolotl in development. The dataset is featured with over 1 million single cells across primary tissues in neotenic and metamorphosed axolotls. They characterized cell-type-specific gene signatures and analyzed dynamic gene expression patterns during limb development. The dataset could be helpful for exploring the molecular identity of cells involved in axolotl development. There are several major concerns, especially about the quality of the dataset, as discussed below.

- 1. The technique uses a very similar cell fixation and barcoding workflow as the published single-cell RNA-seq techniques by combinatorial indexing. Also, it is not obvious to see much improvement in throughput, efficiency, or any new information that can be recovered from the strategy. It is more like an optimized version of the current techniques instead of a new strategy as proposed in the abstract.*

Response: We thank reviewer for this comment. In fact, our method is based on the high-throughput pool-split strategy in sci-RNA-seq (Cao, Packer et al. 2017) and SPLiT-seq (Rosenberg, Roco et al. 2018). We wanted to take advantage of DNA oligo hybridization to expand the barcode combinations of pool-split strategy without using any ligase and ligation step to simplify the protocol and reduce the cost. The major difference between our strategy and published methods is that we depend on DNA oligo hybridization to label the cells rather than ligation. We integrate the ligation in the final step in library construction. On the other hand, most reported pool-split strategy (sci-RNA-

seq, sci-RNA-seq3, SPLiT-seq) were applied to fetal or embryo samples, we also demonstrate the feasibility of pool-split single-cell RNA-seq strategy on multiple adult tissue samples, while other methods (sci-RNA-seq1 and 3; SPLiT-seq) were focused on embryonic or fetal samples. Thus, our strategy could be regarded as another pool-split method for high-throughput single-cell RNA-seq, and could also be extend to other omics. We have expanded the method and performed proof of principle single-cell ATAC-seq experiment on multiple adult tissue samples in other study.

2. *Fig. 1C. For comparing different techniques, the authors should sample the same number of reads per cell. Also, it is not convincing to compare the signal from single-cell RNA-seq with single-nucleus RNA-seq.*

Response: We thank the reviewer for this suggestion. In the revised Figure 1C, we have sampled the same number of reads per cell in all the methods. We also excluded single-nucleus RNA-seq method.

3. *Based on Fig. S1C, there is a strong batch effect between different individuals in both neotenic and metamorphosed axolotls. This batch effect should be removed before downstream analysis.*

Response: We thank the reviewer for this comment. The batch effect between different individuals in both neotenic and metamorphosed axolotls could be introduced in the tissue dissociation step. To evaluate and remove the batch effect, the merged gene expression matrix was processed by the “SCTransform” function in Seurat (Satija, Farrell et al. 2015). We used Scanpy to generate cell clusters for processed data (Wolf, Angerer et al. 2018). We adjusted components and resolutions to generate representative clusters in merged dataset. In the revised version, we removed most batch effect in neotenic axolotl and metamorphosed axolotl datasets before further downstream analyses (Figure S1).

4. *It is critical to ensure that the batch effect does not interfere with the downstream sub-clustering analysis.*

Response: We checked whether certain cell types in tissues were affected by batch effect.

Figure 1 for Reviewer 3. UMAP plots showing the batch effect in major remodeled tissues, certain cell types involved in perturbation were labeled.

As shown in figure1, we discussed some perturbed cell types in line 227-278. Major cell sub-clusters in each tissue which were merged by cells from different individuals were not affected by the batch effect. In some cases, limited number of cells detected in a tissue introduced batch effect of several sub-clusters that were difficult to be integrated into other clusters. The batch effect of these sub-clusters was normalized in differentially expressed genes analysis and gene regulatory network construction. Thus, perturbed cell type analysis, differentially expressed genes analysis and gene regulation analysis were not affected by these sub-clusters.

4. Page 7, line 141: *“Approximately 20% of cells in the library ultimately passed filtration steps”. This is a concern about the quality of the dataset. Why are 80% of cells lost during the filtration step?*

Response: We are sorry that the description in this place is not clear. In the revised version, we modify the words in line 160-162. Around 20% cells entering the experiment were ultimately profiled. In the experiment, pool-split protocols included many centrifugal steps. In the final round of pool-split step, we recovered about 40% cells and counted them. Those 60% lost cells were largely adhered on the inner wall of 96-well plates or lost in the centrifugal steps. We counted and put 5,000 cells in each sub-library with one sequencing index. All the recovered cells in one experiment were divided into 96 sub-libraries. Due to the limitation of sequencing depth, we are not able to profile all the cells in each sub-library. Around 2,000 cells finally passed data quality control with a shallow sequencing depth. Those filtration steps in experiment and data processing lead to a relatively low recovery ratio (~20%) of input cells.

5. It seems some clusters overlap with each other based on the UMAP plot in the sub-cluster analysis (e.g., WT_Gill) but are assigned to different names. This should be clarified.

Response: Based on the UMAP plot, cells with similar expression signatures will be clustered together in dimensional reduction step. Clusters with similar expression patterns will be assigned into neighboring two-dimensional space. Cells in each tissue were collected from all sub-libraries. After data normalization, we chose the same number of principle components and resolution in clustering step of all the tissues. In this case, UMAP plot demonstrated a clear separation in tissues with appropriate number of cells. We observed that in tissues with a large number of cells (especially epithelial cells, e.g. limbs, tail, gill, skin, intestine), some clusters were overlapped due to insufficient differentially expressed genes. We observed heterogeneity in these overlapped spaces. But limited sequencing depth resulted some low-resolution space. We made unbiased cell type annotation for each tissue based on markers generated by same parameter. Actually, most overlapped clusters with different names were assigned to the same cell type. We labeled these clusters with underlines and marker names to distinguish them. The heterogeneity in these overlapped spaces will provide more meaningful information of epithelial cells and stromal cells if the number of differentially expressed genes is enough.

6. Line 289. "Umod was downregulated in metamorphosed axolotl skin." This conclusion is not apparent based on the plot.

Response: We are sorry for our mistake. Umod was enriched in only a small cluster of neotenic axolotl skin cells and showed low expression level in metamorphosed axolotl skin. The description is not representative. We have removed the related plot and claims.

7. The authors claimed that *Chga*⁺ cells were detected only in the neotenic heart. However, this could be simply due to the higher number of cells profiled in the neotenic heart.

Response: We thank reviewer for this comment. Indeed, the number of cells in neotenic heart is much higher than that in metamorphosed heart. Unlike other genes, we could not observe any expression of *Chga* in metamorphosed heart single-cell data (Figure 3G, right, no plot means no expression). On RNA in situ hybridization sections, signals of *Chga* in neotenic heart demonstrate unique expression pattern. We change the claim here: “Interestingly, *Chga* was uniquely expressed in neotenic axolotl heart endocrine cells.” (Line 344–347). We have avoided to use words such as “only” to tone down our claims.

8. Several claims in the manuscript lack support from figures (e.g., the conclusion in line 187, line 202). These should be fixed together with some obvious grammar errors across the manuscript.

Response: We thank the reviewer for reminding us to these mistakes. We have included a new main figure (Figure 3) to support the claims in this part (Line 227–278). We performed RNA *in situ* hybridization on major perturbed genes identified in single-cell datasets. And we also compared other bulk RNA-seq studies and correlated these differentially expressed transcripts with certain perturbed cell types (Bryant, Johnson et al. 2017, Caballero-Perez, Espinal-Centeno et al. 2018, Janet, Juan et al. 2020). In the revised version, we have fixed grammar errors as required.

Bryant, D. M., K. Johnson, T. DiTommaso, T. Tickle, M. B. Couger, D. Payzin-Dogru, T. J. Lee, N. D. Leigh, T. H. Kuo, F. G. Davis, J. Bateman, S. Bryant, A. R. Guzikowski, S. L. Tsai, S. Coyne, W. W. Ye, R. M. Freeman, L. Peshkin, C. J. Tabin, A. Regev, B. J. Haas and J. L. Whited (2017). “A Tissue-Mapped Axolotl De Novo Transcriptome Enables Identification of Limb Regeneration Factors.” Cell Reports **18**(3): 762–776.

Caballero-Perez, J., A. Espinal-Centeno, F. Falcon, L. F. Garcia-Ortega, E. Curiel-Quesada, A. Cruz-Hernandez, L. Bako, X. M. Chen, O. Martinez, M. A. Arteaga-Vazquez, L. Herrera-Estrella and A. Cruz-Ramirez (2018). “Transcriptional landscapes of Axolotl (*Ambystoma mexicanum*).” Developmental Biology **433**(2): 227–239.

Cadiz, L. and M. G. Jonz (2020). “A comparative perspective on lung and gill regeneration.” Journal of Experimental Biology **223**(19).

Cao, J. Y., J. S. Packer, V. Ramani, D. A. Cusanovich, C. Huynh, R. Daza, X. Qiu, C. Lee, S. N. Furlan, F. J. Steemers, A. Adey, R. H. Waterston, C. Trapnell and J. Shendure (2017). “Comprehensive single-cell transcriptional profiling of a multicellular organism.” Science **357**(6352): 661–667.

Herroon, M. K., E. Rajagurubandara, D. L. Rudy, A. Chalasani, A. L. Hardaway and I.

Podgorski (2013). "Macrophage cathepsin K promotes prostate tumor progression in bone." Oncogene **32**(12): 1580-1593.

Janet, P. M., C. P. Juan, E. C. Annie, M. C. Gilberto, L. Hilda, S. V. Enrique, S. Denhi, C. M. Jesus and C. R. Alfredo (2020). "Multi-organ transcriptomic landscape of *Ambystoma velasci* metamorphosis." Developmental Biology **466**(1-2): 22-35.

Jensen, T. B., P. Giunta, N. G. Schulz, Y. Kyeremateng, H. Wong, A. Adesina and J. R. Monaghan (2018). "Neuregulin-1 exerts molecular control over axolotl lung regeneration through ErbB family receptors." bioRxiv: 258517.

Krausgruber, T., N. Fortelny, V. Fife-Gernedl, M. Senekowitsch, L. C. Schuster, A. Lercher, A. Nemc, C. Schmidl, A. F. Rendeiro, A. Bergthaler and C. Bock (2020). "Structural cells are key regulators of organ-specific immune responses." Nature **583**(7815): 296-+.

Rosenberg, A. B., C. M. Roco, R. A. Muscat, A. Kuchina, P. Sample, Z. Z. Yao, L. T. Graybuck, D. J. Peeler, S. Mukherjee, W. Chen, S. H. Pun, D. L. Sellers, B. Tasic and G. Seelig (2018). "Single-cell profiling of the developing mouse brain and spinal cord with split-pool barcoding." Science **360**(6385): 176-+.

Satija, R., J. A. Farrell, D. Gennert, A. F. Schier and A. Regev (2015). "Spatial reconstruction of single-cell gene expression data." Nat Biotechnol **33**(5): 495-502.

Sibai, M., E. Altuntas, B. E. Suzek, B. Sahin, C. Parlayan, G. Ozturk, A. T. Baykal and T. Demircan (2020). "Comparison of protein expression profile of limb regeneration between neotenic and metamorphic axolotl." Biochemical and Biophysical Research Communications **522**(2): 428-434.

Wolf, F. A., P. Angerer and F. J. Theis (2018). "SCANPY: large-scale single-cell gene expression data analysis." Genome Biol **19**(1): 15.

REVIEWERS' COMMENTS

Reviewer #1 (Remarks to the Author):

The availability of a multi-organ, metamorphosis versus neotenic dataset is a new contribution to the field. The authors have addressed a number of the comments from the reviewers. They are still imprecise when coming to describe the biology/physiology of axolotl. :

"The metamorphosis of axolotls eventually results in a severely reduced lifespan. The relatively long lifespan of neotenic axolotls is partially attributed to their extraordinary regenerative capacity8. "

Here they cite a review rather than primary literature--a better referencing of the data regarding lifespan, metamorphosis and regeneration would be needed to include such a statement in the paper.

"The majority of organs in adult axolotls have regenerative capacity, and the systematic cell composition and interaction landscape of axolotls remain to be solved. "
While this statement is toned down, it still is a rather casual statement without documentation or listing of the specific organs.

Reviewer #3 (Remarks to the Author):

The authors have satisfactorily resolved all my comments. I do not have any further concerns. The manuscript should be accepted for publication.

REVIEWERS' COMMENTS

Reviewer #1 (Remarks to the Author):

The availability of a multi-organ, metamorphosis versus neotenic dataset is a new contribution to the field. The authors have addressed a number of the comments from the reviewers.

They are still imprecise when coming to describe the biology/physiology of axolotl. :

"The metamorphosis of axolotls eventually results in a severely reduced lifespan. The relatively long lifespan of neotenic axolotls is partially attributed to their extraordinary regenerative capacity⁸. "

Here they cite a review rather than primary literature--a better referencing of the data regarding lifespan, metamorphosis and regeneration would be needed to include such a statement in the paper.

"The majority of organs in adult axolotls have regenerative capacity, and the systematic cell composition and interaction landscape of axolotls remain to be solved. "

While this statement is toned down, it still is a rather casual statement without documentation or listing of the specific organs.

Reviewer #3 (Remarks to the Author):

The authors have satisfactorily resolved all my comments. I do not have any further concerns. The manuscript should be accepted for publication.

Responses to the reviewers' comments (NCOMMS-21-39949)

Below is our point-by-point response to reviewers' comments. Our responses are in BLUE.

Point-by-point response

Reviewer #1

1. *"The metamorphosis of axolotls eventually results in a severely reduced lifespan. The relatively long lifespan of neotenic axolotls is partially attributed to their extraordinary regenerative capacity⁸. "*

Here they cite a review rather than primary literature--a better referencing of the data regarding lifespan, metamorphosis and regeneration would be needed to include such a statement in the paper.

Response: We thank the reviewer for raising this point. In this part, we mean to introduce the perturbation of regeneration and lifespan in metamorphosed adult axolotl. In revised version, we have cited other primary literatures to support the statement^{1, 2}.

1. Monaghan JR, *et al.* Experimentally induced metamorphosis in axolotls reduces regenerative rate and fidelity. *Regeneration* **1**, 2-14 (2014).
2. Sousounis K, *et al.* A robust transcriptional program in newts undergoing multiple events of lens regeneration throughout their lifespan. *Elife* **4**, (2015).
2. *"The majority of organs in adult axolotls have regenerative capacity, and the systematic cell composition and interaction landscape of axolotls remain to be solved. "*
While this statement is toned down, it still is a rather casual statement without documentation or listing of the specific organs.

Response: In the revised manuscript, we have listed specific organs in line 59-61 (We cited a representative review due to the limited number of references). We deleted the unnecessary duplicate statement ("*The majority of organs in adult axolotls have regenerative capacity*") in this place.